# R-spondins engage heparan sulfate proteoglycans to potentiate WNT signaling

Ramin Dubey[1†], Peter van Kerkhof[2], Ingrid Jordens[2], Tomas Malinauskas[3], Ganesh V Pusapati[1], Joseph K McKenna[4], Dan Li[5], Jan E Carette[6], Mitchell Ho[5], Christian Siebold[3], Madelon Maurice[2], Andres M Lebensohn[4†*], Rajat Rohatgi[1*]

[1]Departments of Biochemistry and Medicine, Stanford University School of Medicine, Stanford, United States; [2]Department of Cell Biology and Oncode Institute, Centre for Molecular Medicine, University Medical Centre Utrecht, Utrecht, Netherlands; [3]Division of Structural Biology, Wellcome Centre for Human Genetics, University of Oxford, Oxford, United Kingdom; [4]Laboratory of Cellular and Molecular Biology, Center for Cancer Research, National Cancer Institute, National Institutes of Health, Bethesda, United States; [5]Laboratory of Molecular Biology, Center for Cancer Research, National Cancer Institute, National Institutes of Health, Bethesda, United States; [6]Department of Microbiology and Immunology, Stanford University School of Medicine, Stanford, United States

*For correspondence:
andres.lebensohn@nih.gov (AML);
rrohatgi@stanford.edu (RR)

†These authors contributed equally to this work

Competing interests: The authors declare that no competing interests exist.

**Abstract** R-spondins (RSPOs) amplify WNT signaling during development and regenerative responses. We previously demonstrated that RSPOs 2 and 3 potentiate WNT/β-catenin signaling in cells lacking leucine-rich repeat-containing G-protein coupled receptors (LGRs) 4, 5 and 6 (Lebensohn and Rohatgi, 2018). We now show that heparan sulfate proteoglycans (HSPGs) act as alternative co-receptors for RSPO3 using a combination of ligand mutagenesis and ligand engineering. Mutations in RSPO3 residues predicted to contact HSPGs impair its signaling capacity. Conversely, the HSPG-binding domains of RSPO3 can be entirely replaced with an antibody that recognizes heparan sulfate (HS) chains attached to multiple HSPGs without diminishing WNT-potentiating activity in cultured cells and intestinal organoids. A genome-wide screen for mediators of RSPO3 signaling in cells lacking LGRs 4, 5 and 6 failed to reveal other receptors. We conclude that HSPGs are RSPO co-receptors that potentiate WNT signaling in the presence and absence of LGRs.

## Introduction

R-spondins (RSPOs) are vertebrate-specific secreted proteins that amplify signaling through the WNT/β-catenin pathway, a cell-cell communication system that regulates tissue patterning during embryonic development and regenerative responses in adults (*de Lau et al., 2014*; *de Lau et al., 2012*; *Nusse and Clevers, 2017*; *Steinhart and Angers, 2018*). Mutations that disrupt the RSPO signaling circuit cause structural birth defects, exemplified by limb truncations, lung hypoplasia, craniofacial malformations and cardiovascular defects seen in humans and mice carrying mutations in *RSPO2* and *RSPO3* (*Aoki et al., 2008*; *Aoki et al., 2007*; *Bell et al., 2008*; *Nam et al., 2007*; *Szenker-Ravi et al., 2018*). In adults, RSPOs function as niche-derived signals required for the renewal of epithelial stem cells in multiple tissues, including the intestine, skin and bone (*de Lau et al., 2014*). Elucidating the mechanisms that mediate the reception and transduction of RSPO signals will further our understanding of these fundamental developmental and homeostatic processes,

and is essential to harnessing the considerable therapeutic potential of this pathway to enhance regenerative responses.

The prevailing paradigm of RSPO signaling holds that they amplify responses to WNT ligands by simultaneously binding to two types of receptors at the cell surface: one of three leucine-rich repeat-containing G-protein coupled receptors (LGRs), LGRs 4, 5 or 6 (hereafter referred to jointly as LGRs), and one of two transmembrane E3 ubiquitin ligases, ZNRF3 or RNF43 (*Carmon et al., 2011*; *Chen et al., 2013*; *de Lau et al., 2011*; *Glinka et al., 2011*; *Gong et al., 2012*; *Peng et al., 2013a*; *Peng et al., 2013b*; *Ruffner et al., 2012*; *Wang et al., 2013*; *Zebisch et al., 2013*; *Zebisch and Jones, 2015*). ZNRF3/RNF43 reduce cell-surface levels of WNT receptors, Frizzled (FZD) and LDL Receptor Related Protein (LRP) 5/6 proteins, by direct ubiquitination (*Hao et al., 2012*; *Koo et al., 2012*). RSPOs amplify WNT signaling by inducing the endocytosis of ZNRF3/RNF43, which leads to an increase in the number of WNT receptors at the plasma membrane and consequently an increase in sensitivity to WNT ligands. RSPOs were thought to increase WNT receptor levels only in specific cell types that express LGRs, such as intestinal stem cells, thus allowing the strength of the response to WNT to be tightly controlled in time and space (*de Lau et al., 2014*).

We and others recently made the unexpected discovery that this paradigm is incomplete: RSPO2 and RSPO3 can potentiate WNT signaling in the absence of LGRs 4, 5 and 6, previously considered obligate high-affinity receptors for RSPOs. Our previous article in *eLife* demonstrated that RSPO2 and RSPO3 can still potentiate WNT signaling in LGR4/5/6$^{KO}$ human haploid cells (*Lebensohn and Rohatgi, 2018*). Similarly, RSPO2 and RSPO3 can amplify WNT-mediated signaling in *Lgr4/5/6* triple-knockout immortalized fibroblasts (*Szenker-Ravi et al., 2018*) and RSPO2 can potentiate WNT signaling in 293T cells lacking LGR4 (*Park et al., 2018*). This 'LGR-independent' mode of signaling is particularly relevant to RSPO-related birth defects because it appears to be the dominant mode of signaling during development of the limbs, lungs and cardiovascular system (*Szenker-Ravi et al., 2018*). Mice carrying loss-of-function mutations in LGRs 4, 5 and 6 (*Lgr4/5/6$^{-/-}$*) do not show either the limb truncations, lung hypoplasia or craniofacial abnormalities observed in humans and mouse embryos lacking *Rspo2* (*Aoki et al., 2008*; *Bell et al., 2008*; *Szenker-Ravi et al., 2018*), or the vascular and associated placental defects observed in mouse embryos lacking *Rspo3* (*Aoki et al., 2007*).

These results raise a pressing question: are there alternative receptors that transduce RSPO signals in the absence of LGRs? RSPOs contain tandem furin-like domains, FU1 and FU2, that interact with ZNRF3/RNF43 and LGRs, respectively. They also contain a thrombospondin type 1 (TSP) domain and a basic region (BR) (we refer to them together as the TSP/BR domain) that interact with heparin and heparan sulfate proteoglycans (HSPGs) (*Ayadi, 2008*; *Chang et al., 2016*; *Nam et al., 2006*; *Ohkawara et al., 2011*; *Ren et al., 2018*). The TSP/BR domain has been considered dispensable to potentiate WNT/β-catenin signaling (*Glinka et al., 2011*; *Kazanskaya et al., 2004*; *Kim et al., 2008*), but we previously demonstrated that it is essential in the absence of LGRs (*Lebensohn and Rohatgi, 2018*). We also proposed that glypicans (GPCs), syndecans (SDCs) or other HSPGs may mediate LGR-independent signaling through interactions between their heparan sulfate (HS) chains and the TSP/BR domain of certain RSPOs.

In this study, we introduced structure-based mutations into the TSP/BR domain of RSPO3, engineered RSPO3 ligands fused to a non-native, HSPG-binding single-chain variable fragment (scFv) and disrupted different classes of HSPGs to demonstrate that the interaction between RSPO3 and the HS chains of HSPGs is necessary and sufficient to transduce signals in cells lacking LGRs. We find that the interaction with HSPGs also contributes substantially to LGR-dependent signaling in cultured cells and in small intestinal organoids, showing that HSPGs play an important role in promoting RSPO3 signaling in most contexts.

# Results

## The interaction between the TSP/BR domain of RSPO3 and HSPGs is necessary to potentiate WNT signaling

We previously reported that disruption of EXTL3, the gene encoding a glycosyltransferase required for the biosynthesis of all HS chains, reduces LGR-independent RSPO3 signaling by ~80% (*Lebensohn and Rohatgi, 2018*). However, depleting either all SDCs (1-4) or PIGL, an enzyme required for the biogenesis of all GPCs, did not reduce LGR-independent signaling by RSPO3. These observations suggested functional redundancy between GPCs, SDCs and perhaps other HSPGs (*Sarrazin et al., 2011*) in mediating LGR-independent signaling, making loss-of-function genetic experiments aimed at characterizing this signaling mechanism challenging. Our previous experiments also left open the possibility that reducing the levels of all HSPGs by depleting EXTL3 may change the cell-surface abundance of a different co-receptor for RSPOs that mediates LGR-independent signaling. Therefore, we investigated the mechanism of LGR-independent signaling further by modifying the RSPO ligands themselves.

We previously showed that deleting the complete TSP/BR domain of RSPO3 eliminated its ability to potentiate WNT signaling in LGR4/5/6$^{KO}$ cells (*Lebensohn and Rohatgi, 2018*). Deleting the entire TSP/BR domain is a drastic alteration that prevents binding to heparin (*Nam et al., 2006*), but could also disrupt interactions with other potential binding partners. To design more precise mutations that selectively disrupt the putative HS-binding surface of RSPO3, we created a structural model of the RSPO3 TSP/BR domain based on its homology to 26 of the most similar proteins in the Protein Data Bank (*Figure 1A* and *Figure 1—video 1*). Analogous to other TSP domains, the model shows an elongated three-stranded fold stabilized by disulfide bonds at both ends (*Ayadi, 2008*; *Tan et al., 2002*). Two positively charged grooves (Site-1 and Site-2 in *Figure 1A* and *Figure 1—video 1*) on one face of the model suggested potential sites of interaction with negatively charged HS polymers.

Based on this model, we mutated six (out of sixteen) lysine (K) and four (out of nine) arginine (R) residues in these grooves to charge-reversing glutamic acid (E) residues, thereby altering the electrostatic surface of the TSP/BR domain so that it should not be able to interact properly with HS chains. We refer to this mutant protein as RSPO3 TSP/BR (K/R→E) (*Figure 1A and B*, *Figure 1—video 1*, and *Supplementary file 1*). We produced a tagged version of RSPO3 TSP/BR (K/R→E) and purified it under physiological conditions using a one-step affinity purification strategy described in our previous *eLife* paper (see Figure 1—figure supplement 1A from *Lebensohn and Rohatgi, 2018*). We have demonstrated that tagged RSPO proteins produced and purified in this manner preserve the signaling properties of their untagged counterparts (see Figure 1—figure supplement 1C and D from *Lebensohn and Rohatgi, 2018*). A coomassie blue-stained gel of purified wild type (WT) RSPO3, RSPO3 TSP/BR (K/R→E) and RSPO3 ΔTSP/BR, a control construct lacking the TSP/BR domain entirely (*Figure 1B*), is shown in *Figure 1C*.

We tested the capacity of these recombinant RSPO3 proteins to potentiate signaling induced by a low dose of WNT3A conditioned medium (CM) in WT HAP1-7TGP or in LGR4/5/6$^{KO}$ cells. HAP1-7TGP is a human haploid cell line harboring a fluorescent transcriptional reporter for WNT/β-catenin signaling that we have previously established as a valid and genetically tractable system to study both WNT- and RSPO-dependent signaling (*Lebensohn et al., 2016*). LGR4/5/6$^{KO}$ cells are a clonal derivative of HAP1-7TGP in which LGRs 4, 5 and 6 were disrupted by CRISPR/Cas9-mediated genome editing (*Lebensohn and Rohatgi, 2018*). In LGR4/5/6$^{KO}$ cells, the signaling response to low levels of WNT3A can be potentiated robustly by RSPO2 and RSPO3, but not by RSPO1 or RSPO4 (see Figure 1B and C from *Lebensohn and Rohatgi, 2018*), making them a convenient system to study LGR-independent RSPO signaling.

As we had observed previously (see Figure 2F and G from *Lebensohn and Rohatgi, 2018*), deleting the TSP/BR domain of RSPO3 reduced its potency by 82-fold in WT HAP1-7TGP cells (*Figure 1D*), and completely abolished signaling in LGR4/5/6$^{KO}$ cells (*Figure 1E*). Similarly, the K/R→E mutations we introduced into the TSP/BR domain of RSPO3 reduced its potency by 13-fold in WT HAP1-7TGP cells (*Figure 1D*) and almost completely abolished its ability to potentiate WNT signaling in LGR4/5/6$^{KO}$ cells (*Figure 1E*). Our data supports the model that interactions between basic K and R residues in the TSP/BR domain of RSPO3 and the HS chains of HSPGs contribute

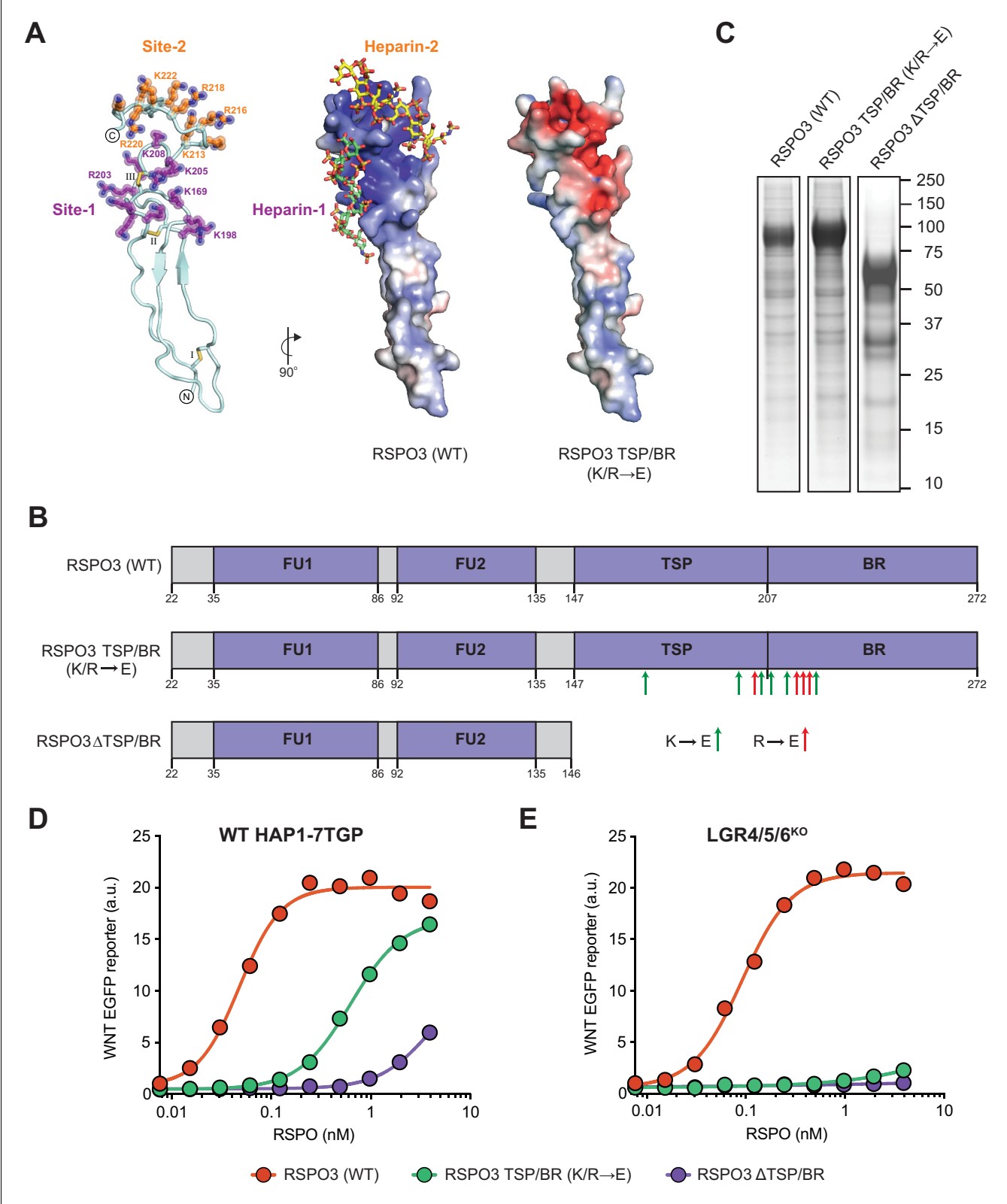

**Figure 1.** Mutations in the putative HSPG-binding surface of the RSPO3 TSP/BR domain impair its ability to amplify WNT signaling. (**A**) Homology model of the isolated TSP/BR domain of human RSPO3 (residues valine 146 - isoleucine 232; UniprotKB Q9BXY4). Left panel: cartoon representation of the protein backbone, with side-chains shown for lysine (K) and arginine (R) residues. N- and C-termini are circled and disulfide bonds are labelled with roman numerals. Two positively charged HS-binding grooves, revealed by the electrostatic potential map (middle panel), are labeled Site-1 and Site-2.

*Figure 1 continued on next page*

*Figure 1 continued*

K and R residues lining Site-1 and Site-2 are colored purple and orange, respectively, and residues mutated to glutamic acid (E) in the RSPO3 TSP/BR (K/R→E) mutant are labeled. Middle panel: the electrostatic potential map calculated with the Adaptive Poisson-Boltzmann Solver (*Jurrus et al., 2018*) is displayed from −10 kT/e to +10 kT/e (blue: positively charged; red: negatively charged). Two virtually docked 8-mer heparin chains (estimated free energies are −13.9 and −14.8 kcal/mol for Site-1 and Site-2, respectively) are depicted in stick representation (red: oxygen, blue: nitrogen, gold: sulphur, green: carbon of heparin-1, yellow: carbon of heparin-2). Right panel: electrostatic potential map of the RSPO3 TSP/BR (K/R→E) mutant. The orientation of the middle and right panels is 90° rotated clockwise around the y-axis relative to the left panel. (B) Schematic representation of WT and mutant RSPO3 variants showing the domains and mutations present in these proteins. The N-terminal HA and the C-terminal Fc and 1D4 tags present in these constructs are not shown. Amino acid numbers for human RSPO3 (UniProtKB Q9BXY4) are indicated below. Green and red arrows show K→E and R→E mutations, respectively, introduced into the TSP/BR domain of the RSPO3 TSP/BR (K/R→E) mutant. Polypeptide lengths are drawn to scale. See *Supplementary file 1* for the nucleotide sequences of these constructs. (C) Coomassie-stained polyacrylamide gels showing equal volumes of the three purified RSPO3 proteins used in (D) and (E). Molecular weight standards in kilodaltons (kDa) are indicated to the right. (D) and (E) Dose-response curves for the indicated purified RSPO3 variants in WT HAP1-7TGP (D) and LGR4/5/6$^{KO}$ (E) cells, in the presence of 1.43% WNT3A CM. Each circle represents the median WNT reporter fluorescence from 2,500 cells. Dose-response curves were fitted to the data using non-linear regression as described in Materials and methods. Where possible, half-maximal effective concentrations (EC$_{50}$, all in nM) were derived from curve fits and are as follows. RSPO3 (WT): 0.048 ± 0.005 in (D) and 0.092 ± 0.007 in (E). RSPO3 ΔTSP/BR (K/R→E): 0.63 ± 0.02 in (D). RSPO3 ΔTSP/BR: 3.93 ± 1.0 in (D).

The online version of this article includes the following video for figure 1:

**Figure 1—video 1.** Movie showing a structural homology model of the WT RSPO3 TSP/BR domain and the K/R→E mutant.
https://elifesciences.org/articles/54469#fig1video1

---

substantially to potentiating WNT signaling in the presence of LGRs. In the absence of LGRs, the RSPO3-HSPG interaction becomes essential to potentiate WNT signaling.

## The TSP/BR domain of RSPO3 can be replaced by a non-native HS-binding domain without compromising activity

While the positively charged grooves we mutated in the TSP/BR domain (*Figure 1A* and *Figure 1—video 1*) are known to interact with HS chains in other TSP domain-containing proteins (*Sarrazin et al., 2011*; *Tan et al., 2002*), it remained possible that the K/R→E mutations disrupted the interaction of RSPO3 with a different, unidentified receptor. To resolve this issue, we sought to provide the HSPG-binding functionality of the TSP/BR domain through an entirely different protein. We engineered a synthetic RSPO3 protein in which we replaced the entire TSP/BR domain with HS20, an scFv antibody selected by phage display to bind to the HS chains of GPC3 (*Gao et al., 2016*; *Gao et al., 2014*; *Geoghegan et al., 2017*; *Figure 2A* and *Supplementary file 1*). We refer to this protein as RSPO3 ΔTSP/BR HS20. Due to the presence of the HS20 scFv in the RSPO3 ΔTSP/BR HS20 fusion protein, we excluded the Fc tag that we had used in other RSPO3 constructs to promote their solubility. We retained an N-terminal HA tag and a small C-terminal 1D4 tag, the latter of which was used for affinity purification of the fusion protein (*Figure 2A and B*).

Replacing the TSP/BR domain with HS20 in RSPO3 ΔTSP/BR HS20 did not disrupt the ability of RSPO3 to signal both in the presence and absence of LGRs (*Figure 2C and D*). Importantly, mutations in the antigen-binding third complementarity-determining region (CDR3) of HS20 (RSPO3 ΔTSP/BR HS20 (GS) and RSPO3 ΔTSP/BR HS20 (A) in *Figure 2A and B*) entirely eliminated the WNT-potentiating activity of RSPO3 ΔTSP/BR HS20, both in the presence and absence of LGRs (*Figure 2E and F*). The fact that HS20, a structurally unrelated protein specifically selected for HS-binding activity (*Gao et al., 2014*), can functionally substitute for the TSP/BR domain of RSPO3 provides strong support for our model that interactions with HSPGs are required to potentiate WNT signaling in cells lacking LGRs, and also contribute significantly in cells containing LGRs.

What are the relative contributions of ZNRF3/RNF43, LGRs and HSPGs to signaling by RSPO3 ΔTSP/BR HS20? In order to measure the contribution of FU1 domain interactions with ZNRF3/RNF43 and FU2 domain interactions with LGRs, we individually mutated the FU1 and FU2 domains of RSPO3 ΔTSP/BR HS20. Point mutations (R67A/Q72A) in the FU1 domain of RSPO3 ΔTSP/BR HS20 (RSPO3 ΔTSP/BR HS20 (R67A/Q72A) in *Figure 2A and B*) known to weaken the interaction with ZNRF3/RNF43 (*Xie et al., 2013*), reduced signaling potency by 10-fold in WT HAP1-7TGP cells (*Figure 2G*) and abolished signaling in LGR4/5/6$^{KO}$ cells (*Figure 2H*). Point mutations (F106E/F110E) in the FU2 domain of RSPO3 ΔTSP/BR HS20 (RSPO3 ΔTSP/BR HS20 (F106E/F110E) in *Figure 2A and B*) that weaken interactions with LGRs (*Xie et al., 2013*) reduced signaling efficacy by 80% and

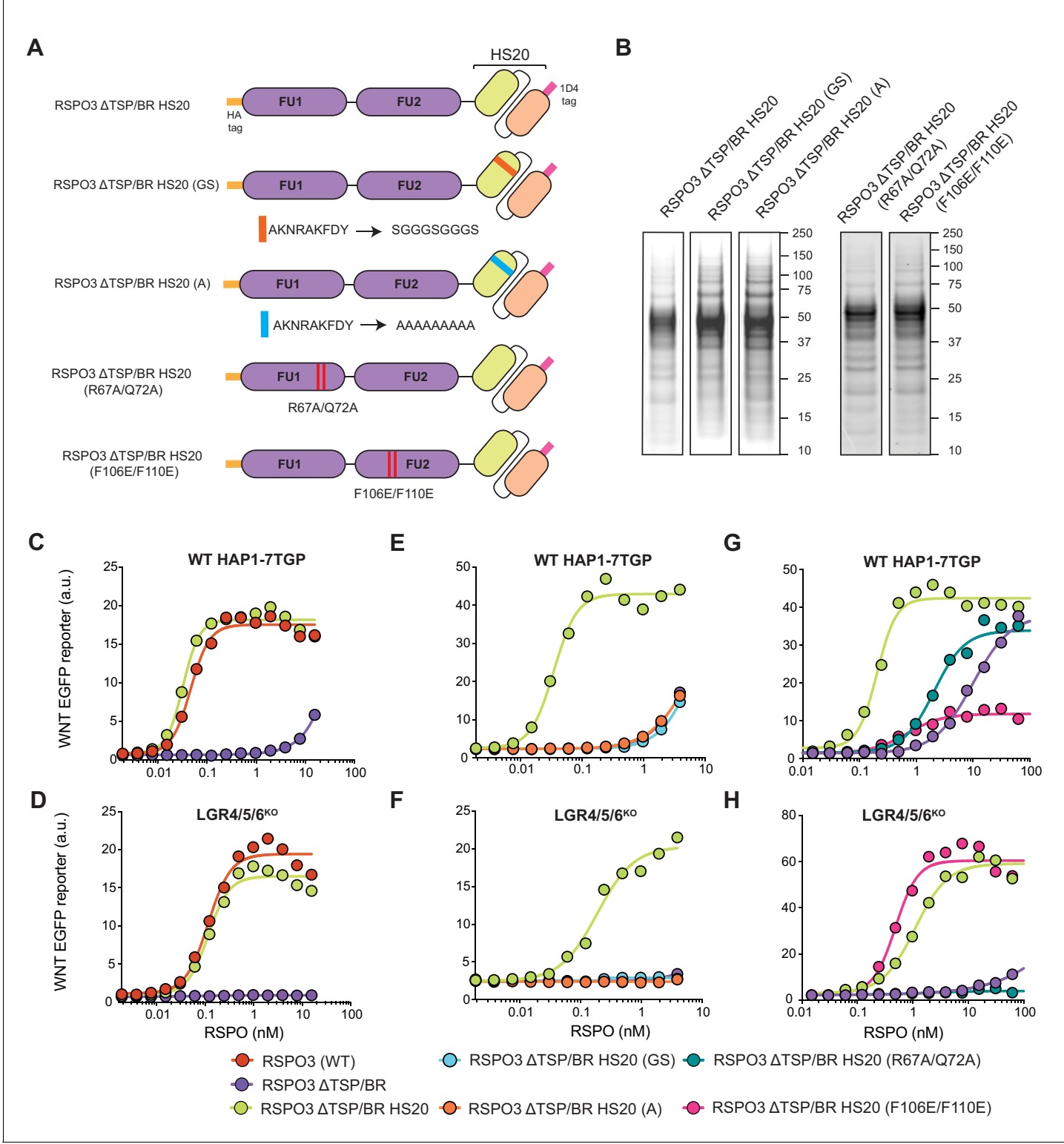

**Figure 2.** The TSP/BR domain of RSPO3 can be replaced by HS20, an scFv that binds HS chains, without compromising activity. (A) Cartoons depicting synthetic proteins in which the FU1 and FU2 domains of RSPO3 were fused to HS20, an scFv that binds to HS chains. In two control proteins designed to abolish HS binding, the CDR3 loop of HS20 was replaced with a glycine-serine (GS) or a poly-alanine (A) linker. In two other proteins, we introduced mutations that diminish binding of the FU1 domain to ZNRF3/RNF43 (R67A/Q72A) or of the FU2 domain to LGRs (F106E/F110E). See *Supplementary file 1* for the nucleotide sequences of these constructs. (B) Coomassie-stained polyacrylamide gels showing equal volumes of the five purified RSPO3 proteins used in (C–H). Molecular weight standards in kDa are indicated to the right of each gel. (C–H) Dose-response curves for the

*Figure 2 continued on next page*

*Figure 2 continued*

indicated purified RSPO3 variants in WT HAP1-7TGP (**C, E and G**) and LGR4/5/6$^{KO}$ (**D, F and H**) cells, in the presence of 1.43% WNT3A CM. Each circle represents the median WNT reporter fluorescence from 2,500 cells. Dose-response curves were fitted to the data using non-linear regression as described in Materials and methods. Where possible, EC$_{50}$ values (all in nM) were derived from curve fits and are as follows. RSPO3 (WT): 0.046 ± 0.004 in (**C**) and 0.11 ± 0.015 in (**D**). RSPO3 ΔTSP/BR: 9.8 ± 0.92 in (**G**). RSPO3 ΔTSP/BR HS20: 0.032 ± 0.002 in (**C**), 0.11 ± 0.013 in (**D**), 0.035 ± 0.003 in (**E**), 0.18 ± 0.03 in (**F**), 0.20 ± 0.017 in (**G**) and 1.1 ± 0.12 in (**H**). RSPO3 ΔTSP/BR HS20 (R67A/Q72A): 2.0 ± 0.26 in (**G**). RSPO3 ΔTSP/BR HS20 (F106E/F110E): 0.68 ± 0.014 in (**G**) and 0.49 ± 0.07 in (**H**).

potency by 3-fold in WT HAP1-7TGP cells (*Figure 2G*). As expected, disrupting the LGR-binding FU2 domain in RSPO3 ΔTSP/BR HS20 (F106E/F110E) did not reduce signaling in LGR4/5/6$^{KO}$ cells (*Figure 2H*).

We conclude that interactions with ZNRF3/RNF43, LGRs and HSPGs all contribute to the potentiation of WNT signaling by RSPO3 ΔTSP/BR HS20 in the presence of LGRs. Interactions with ZNRF3/RNF43 and HSPGs are required for signaling in the absence of LGRs. The simplest model to explain these results is that RSPO3 can bind simultaneously to LGR and HSPG co-receptors, which together present the ligand to ZNRF3/RNF43, the key effectors that mediate endocytosis of the complex. Without LGRs, RSPO3 is presented by HSPGs alone, explaining the significantly lower potency of both WT RSPO3 and RSPO3 ΔTSP/BR HS20 in LGR4/5/6$^{KO}$ compared to WT HAP1-7TGP cells (*Figures 1D, E*, *2C and D*). Binding assays and structural studies with purified proteins will be required to test for the formation of a ternary RSPO3-LGR-HSPG complex.

## Signaling by RSPO3 causes internalization and degradation of RNF43 in the presence or absence of LGRs

We tested whether the mechanism whereby RSPO3 ΔTSP/BR HS20 potentiates WNT responses is equivalent to that of WT RSPO3. LGR-dependent RSPO signaling triggers internalization and lysosomal degradation of the E3 ubiquitin ligases ZNRF3 and RNF43, promoting the stabilization of WNT receptors (*Hao et al., 2012*; *Koo et al., 2012*). Whether LGR-independent signaling by RSPO3 proceeds through a similar mechanism is not known. We used a surface biotinylation assay (*Koo et al., 2012*) to measure RSPO3-mediated internalization and degradation of RNF43-Flag in WT HAP1-7TGP and in LGR4/5/6$^{KO}$ cells. Treatment of either WT HAP1-7TGP or LGR4/5/6$^{KO}$ cells with WT RSPO3 led to the depletion of RNF43-Flag from the cell surface and a reduction in its total cellular level (*Figure 3A and B*), suggesting similar signaling mechanisms for LGR-dependent and LGR-independent potentiation of WNT signaling. These results are consistent with our previous mutational analysis demonstrating that the ZNRF3/RNF43-interacting FU1 domain is essential for RSPO3 signaling in cells lacking LGRs (*Lebensohn and Rohatgi, 2018*).

Having established the mechanism of LGR-independent signaling by RSPO3, we tested whether RSPO3 ΔTSP/BR HS20 potentiates WNT responses through an equivalent mechanism. Deletion of the TSP/BR domain of RSPO3 (RSPO3 ΔTSP/BR in *Figure 3A and B*) prevented the internalization and degradation of RNF43-Flag in both WT HAP1-7TGP and LGR4/5/6$^{KO}$ cells, consistent with the reduced capacity of RSPO3 ΔTSP/BR to promote LGR-dependent (*Figure 1D*) and LGR-independent (*Figure 1E*) signaling. In contrast, as with WT RSPO3, addition of RSPO3 ΔTSP/BR HS20 to either WT HAP1-7TGP (*Figure 3A*) or LGR4/5/6$^{KO}$ (*Figure 3B*) cells resulted in the depletion of RNF43-Flag from the cell surface and a reduction in its total cellular level. The effects of RSPO3 ΔTSP/BR HS20 were dependent on the interaction between HS20 and HS chains, since the RSPO3 ΔTSP/BR HS20 (GS) mutant, which carries mutations in the HS-binding CDR3 loop of HS20, failed to promote internalization and degradation of RNF43-Flag (*Figure 3A and B*).

In summary, replacing the HS-binding TSP/BR domain of RSPO3 with HS20, an unrelated protein that binds HS chains, can rescue signaling in the absence of LGRs and substantially enhance signaling in their presence by promoting the internalization and degradation of RNF43. We propose that HSPGs represent the third major class of RSPO co-receptors, in addition to LGRs and ZNRF3/RNF43.

## HSPG specificity of RSPO3 signaling

Are specific HSPGs required for RSPO signaling? Cell-surface HSPGs comprise six glypicans (GPC1-6), which are anchored to the plasma membrane by a glycophosphatidylinositol (GPI) linkage, and

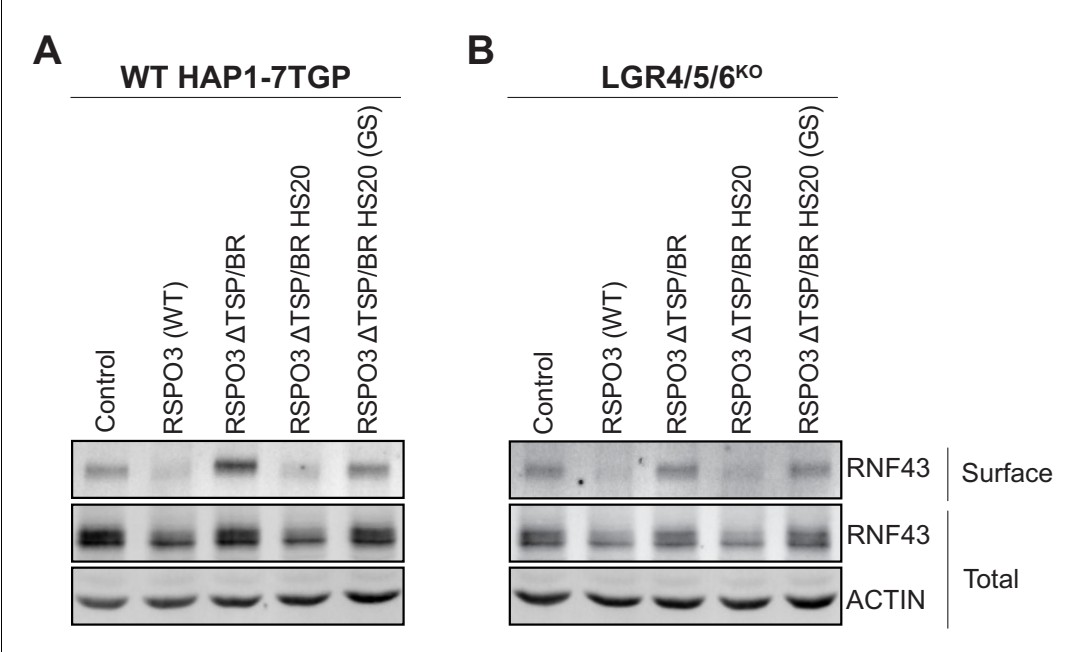

**Figure 3.** Signaling by WT RSPO3 and RSPO3 ΔTSP/BR HS20 causes internalization and degradation of RNF43 in the presence and absence of LGRs. Cell-surface RNF43-Flag (captured by cell-surface biotinylation; see Materials and methods), total RNF43-Flag and total actin were visualized by immunoblotting after treatment of WT HAP1-7TGP (A) or LGR4/5/6^{KO} (B) cells with the indicated RSPO3 variants.

four syndecans (SDC1-4), which are single pass transmembrane proteins (*Sarrazin et al., 2011*). In our previous *eLife* paper (*Lebensohn and Rohatgi, 2018*), we presented evidence that RSPO3 can use the HS chains on multiple HSPGs to potentiate WNT signaling. However, HS20 is an scFv selected to bind to GPC3 (*Gao et al., 2014*), raising the possibility that the synthetic RSPO3 ΔTSP/ BR HS20 ligand may have a narrower HSPG selectivity. In a binding assay with purified proteins, HS20 bound equally well to GPC3 and GPC4 (*Figure 4A*), the two GPCs expressed most highly in HAP1 cells (see Table 1 in *Lebensohn and Rohatgi, 2018*: average RPKM values for GPC3 and GPC4 were 157 and 220, respectively). HS20 failed to bind to mutant GPC3 and GPC4 proteins that cannot be modified by HS chains (GPC3ΔHS and GPC4ΔHS in *Figure 4A*). These results support a prior conclusion that HS20 recognizes the HS side-chain structure shared by multiple HSPGs, rather than binding to the protein core of a specific GPC (*Gao et al., 2014*).

To test the HSPG specificity of RSPO3 ΔTSP/BR HS20 more thoroughly in cells, we used CRISPR/ Cas9 methods to generate a series of clonal cell lines (see Materials and methods and *Supplementary files 2* and *3*) with increasingly severe deficits in their HSPG composition: (1) GPC3^{KO}, (2) GPC4^{KO}, (3) PIGL^{KO} (lacking an enzyme required for GPI anchor biosynthesis and hence for the cell-surface expression of all GPCs), (4) SDC1/2/3/4^{KO} (lacking all four SDCs) and (5) EXTL3^{KO} (lacking a glycosyltransferase required for the synthesis of HS chains attached to all HSPGs, including all GPCs and SDCs, but not for the synthesis of other proteoglycans or glycosaminoglycans). Each of these mutations was introduced into LGR4/5/6^{KO} cells, in which the HS-binding activity of HS20 is required for signaling (*Figure 2D*), and hence the effect of eliminating different HSPGs can be measured most accurately. We also tested the specificity of WT RSPO3, containing the native HSPG-binding TSP/BR domain, in these cell lines.

Disrupting GPC3 in LGR4/5/6^{KO} cells had no significant effect on signaling induced by either WT RSPO3 or RSPO3 ΔTSP/BR HS20 (*Figure 4B and C*), consistent with the finding that HS20 can bind with equivalent affinity to the HS chains of other GPCs such as GPC4 (*Figure 4A*). Disrupting GPC4 or PIGL modestly reduced the potency (i.e. increased the $EC_{50}$) of both WT RSPO3 and RSPO3 ΔTSP/BR HS20 (*Figure 4B and C*). Combined disruption of all SDCs – SDC1, SDC2, SDC3 and SDC4 – surprisingly increased the potency (ie. lowered the $EC_{50}$) of both WT RSPO3 and RSPO3 ΔTSP/BR

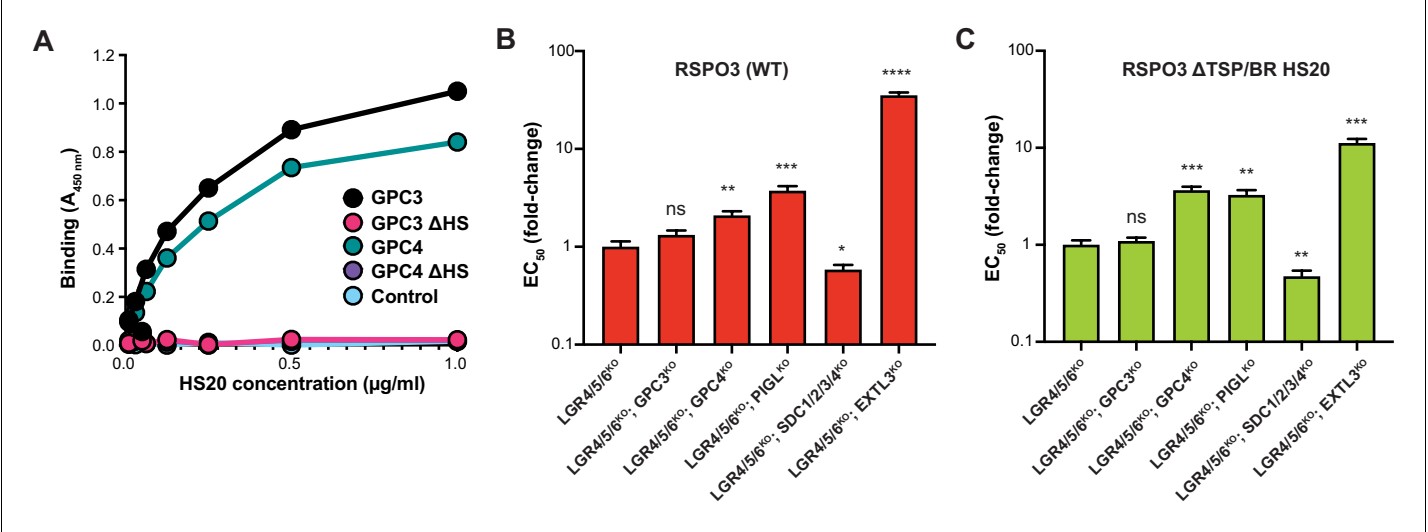

**Figure 4.** RSPO3 ΔTSP/BR HS20 engages the HS chains of multiple HSPGs. (**A**) Binding of HS20 to GPC3 or GPC4. An ELISA plate was coated with GPC3, GPC4, or mutant versions lacking HS chains (GPC3ΔHS or GPC4ΔHS). HS20 scFv was added at increasing concentrations (x-axis), followed by a goat anti-human IgG horseradish peroxidase conjugate. Bound HS20 was quantified with 3,3′,5,5′-tetramethylbenzidine detection reagent by measuring absorbance at 450 nm (y-axis). An irrelevant Fc-fusion protein (CD276-hFc, labeled 'Control') was used as an antigen control. (**B–C**) Fold-change in the $EC_{50}$ of WT RSPO3 (**B**) or RSPO3 ΔTSP/BR HS20 (**C**) in LGR4/5/6$^{KO}$ cells and clonal derivatives in which different HSPGs were depleted through the indicated gene disruptions (see Results and Materials and methods for details). Since deletion of some GPCs in HAP1 cells leads to impaired WNT reception (*Lebensohn et al., 2016*; *Lebensohn and Rohatgi, 2018*), we first determined a concentration of WNT3A CM that yielded similar levels of WNT reporter activity in all the cell lines and then titrated RSPO3 variants to determine their $EC_{50}$ values. Error bars denote the standard error of the curve fits used to calculate the $EC_{50}$. The statistical significance of differences between the measured $EC_{50}$ values in LGR4/5/6$^{KO}$ cells and clonal derivatives thereof was determined by a two-tailed, unpaired t-test, and is indicated as **** ($p<0.0001$), *** ($p<0.001$), ** ($p<0.01$), * ($p<0.05$) or ns (non-significant).

HS20 (*Figure 4B and C*). This increase in potency may be related to the previous observation that some SDCs inhibit WNT/β-catenin signaling through regulation of LRP6 and RSPO3 (*Astudillo et al., 2014*). Finally, disrupting EXTL3, predicted to globally block HS chain biosynthesis, impaired signaling by both proteins to the greatest extent (*Figure 4B and C*).

Overall, disrupting different HSPGs led to a qualitatively similar effect on signaling by WT RSPO3 and RSPO3 ΔTSP/BR HS20, with GPC3 disruption having no effect, GPC4 or PIGL disruption causing a modest reduction in signaling potency and EXTL3 disruption causing the greatest reduction in signaling potency. These results are most consistent with the model that RSPO3 ΔTSP/BR HS20, like WT RSPO3, can signal in a redundant manner via either GPCs, SDCs or another HSPG by engaging their HS chains rather than their protein cores.

### HSPGs potentiate RSPO3-supported small intestinal organoid growth

Given the substantial contribution made by the interaction between the TSP/BR domain or HS20 and HSPGs to LGR-dependent signaling in haploid cells (*Figures 1D* and *2C*), it was important to test whether this interaction also mediates RSPO signaling in a more physiological context. The growth of mouse small intestinal organoids is strictly dependent on exogenously supplied RSPOs (*Sato et al., 2009*) and on the expression of LGRs (*de Lau et al., 2011*). In addition, studies in mice have shown that depletion of HS chains in the intestinal epithelium compromise WNT-dependent crypt renewal and regeneration following radiation-induced injury (*Yamamoto et al., 2013*). We used this organoid culture system in combination with our mutant and engineered RSPO3 ligands to test the importance of RSPO-HSPG interactions for supporting the crypt-villus architecture of the intestine. WT RSPO3 promoted the growth of intestinal organoids with multiple budded structures at concentrations down to 0.5 nM (*Figure 5A and B*). The buds represent crypt domains, since they are enriched in markers of proliferating cells (Ki67) and of Paneth cells (lysozyme) (*Figure 5C*). RSPO3 ΔTSP/BR was markedly less potent in this assay, enabling only a few organoids to grow at a

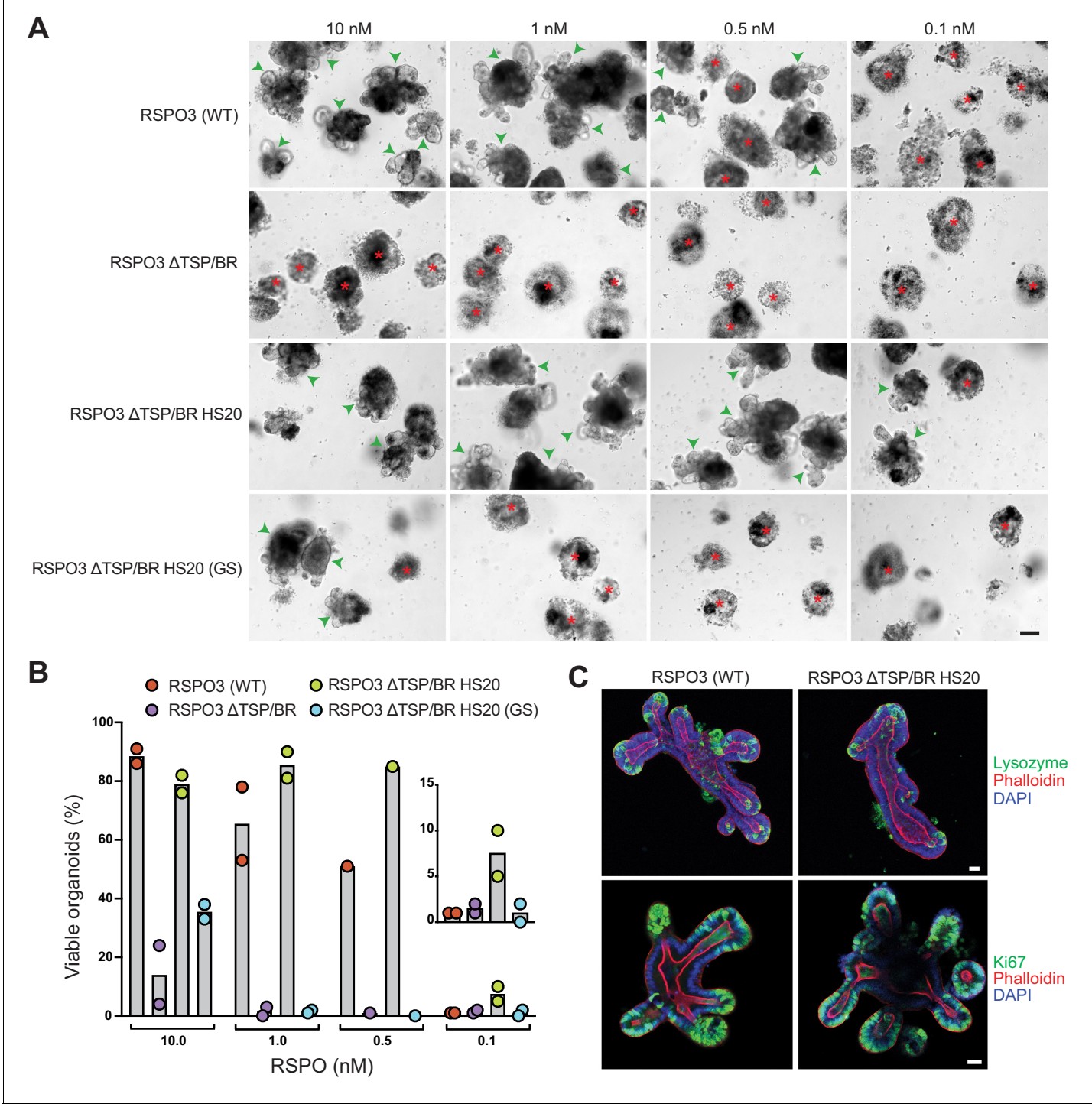

**Figure 5.** The interaction with HSPGs potentiates the ability of RSPO3 to support the growth of small intestinal organoids. (**A**) Bright-field microscopy images of B6 mouse small intestinal organoids grown in EN medium supplemented with various purified RSPO3 proteins at the indicated concentrations. Green arrowheads indicate viable, crypt-containing organoids and red asterisks mark dead organoids. The scale bar in the bottom right image represents 100 μm. (**B**) The viability of organoids grown at various concentrations of the indicated RSPO3 variants was quantified from images of the type shown in (**A**). Each circle represents the quantification of an independent experiment. The inset (right) shows a magnified view of organoid viability at a ligand concentration of 0.1 nM. (**C**) Confocal microscopy images of mouse small intestinal organoids grown in medium supplemented with 1 nM of the indicated purified RSPO3 proteins. DAPI (blue) stains nuclei, phalloidin (red) stains actin filaments at the apical surface of cells, lysozyme (green, top row) stains Paneth cells and Ki67 (green, bottom row) stains proliferating cells. For both images in each row, the scale bars in the right image represent 30 μm.

concentration of 10 nM (*Figure 5B*). While the FU1 and FU2 domains present in RSPO3 ΔTSP/BR were sufficient to promote organoid growth at the highest concentration tested, our titration experiment revealed that abolishing HSPG interactions by deleting the TSP/BR domain sharply reduced ligand potency.

To determine if the positive effect of the TSP/BR domain in promoting organoid growth is due to its interactions with HSPGs, we tested the organoid-supporting properties of RSPO3 ΔTSP/BR HS20. Not only was RSPO3 ΔTSP/BR HS20 able to fully support organoid growth, with budding of normal crypt domains, but it was more potent than WT RSPO3, supporting some organoid growth even at concentrations as low as 0.1 nM (*Figure 5A and B*). Organoids grown in the presence of RSPO3 ΔTSP/BR HS20 were indistinguishable from those grown in the presence of WT RSPO3, with crypt domains containing proliferating cells and Paneth cells (*Figure 5C*). The capacity of RSPO3 ΔTSP/BR HS20 to promote organoid growth required the interaction between HS20 and HS chains, since RSPO3 ΔTSP/BR HS20 (GS), carrying mutations in the HS-binding CDR3 loop (*Figure 2A*) failed to promote organoid growth at concentrations below 10 nM, similarly to RSPO3 ΔTSP/BR (*Figure 5A and B*).

We conclude from these experiments that the interaction between the TSP/BR domain, or HS20, and HSPGs contributes substantially to the capacity of RSPO3 to support the growth of mouse intestinal organoids. While additional work will be required to establish which specific HSPGs mediate RSPO reception in the intestine, we note that *Gpc3* mRNA expression in mouse intestinal organoids is undetectable (*Lindeboom et al., 2018*), suggesting that, as in HAP1 cells (*Figure 4B*), RSPOs likely signal through other HSPGs. Multi-omics data shows that GPC4, SDC1 and SDC4 are expressed at detectable levels in mouse intestinal organoids (*Lindeboom et al., 2018*), making them candidate receptors for RSPO3 in this tissue.

## A haploid genetic screen for RSPO3 receptors in cells lacking LGRs

While our results demonstrate that the interaction between RSPO3 and the HS chains of HSPGs is necessary to promote LGR-independent signaling in cells, and can significantly enhance LGR-dependent signaling in cells and intestinal organoids, they do not preclude the existence of another co-receptor capable of mediating these effects. We previously reported the results of a genome-wide, haploid genetic screen for regulators of RSPO1-potentiated, low-level WNT signaling in WT HAP1-7TGP cells (*Lebensohn et al., 2016*). That screen was focused on LGR-dependent signaling, since RSPO1 (unlike RSPO3) requires LGRs to potentiate WNT signaling (*Lebensohn and Rohatgi, 2018*). We attempted to identify any additional co-receptors required for LGR-independent signaling through a similar haploid genetic screen for regulators of RSPO3-potentiated, low-level WNT signaling in cells lacking LGRs 4, 5 and 6 (see Materials and methods). To distinguish genes required specifically for LGR-independent signaling from those required for LGR-dependent signaling, we compared the results of our new screen to those of our previous screen for RSPO1-potentiated signaling in WT HAP1-7TGP cells (*Lebensohn et al., 2016*).

The RSPO3 screen in LGR4/5/6^KO cells revealed a subset of the WNT signaling regulators previously identified in the RSPO1 screen in WT HAP1-7TGP cells (*Figure 6A–C* and *Supplementary files 4* and *5*), including the WNT co-receptor LRP6, members of the TCF/LEF transcriptional complex such as CTNNB1, CREBBP, BCL9 and DOT1L, the ubiquitin ligase RNF146, which ubiquitinates poly-ADP-ribosylated AXIN1 for proteasomal degradation, and a truncated form of AXIN2 that represses WNT signaling through an unknown mechanism (*Lebensohn et al., 2016*). LGR4, the top hit of the RSPO1 screen in WT HAP1-7TGP cells, was not a significant hit in the RSPO3 screen in LGR4/5/6^KO cells, in which LGR4 had been disrupted by CRISPR/Cas9-mediated genome editing (*Figure 6A–C* and *Supplementary files 4* and *5*).

The RSPO3 screen in LGR4/5/6^KO cells failed to reveal any unique significant hits compared to the RSPO1 screen in WT HAP1-7TGP cells (*Figure 6A–C* and *Supplementary files 4* and *5*), suggesting there were no genes selectively required for LGR-independent potentiation of WNT signaling by RSPO3. While several components of the GPI-anchor biosynthetic machinery, including PIGP, PIGS, PIGU, PIGL and DPM3, as well as GPC4 were hits in both screens, we have previously shown that their presence as significant hits in the RSPO1 screen in WT HAP1-7TGP cells was due to the requirement of GPI-anchored GPCs for low-level WNT signaling, independently of RSPOs (*Lebensohn et al., 2016*). In summary, haploid genetic screens did not reveal an additional co-receptor required for LGR-independent potentiation of WNT signaling by RSPO3.

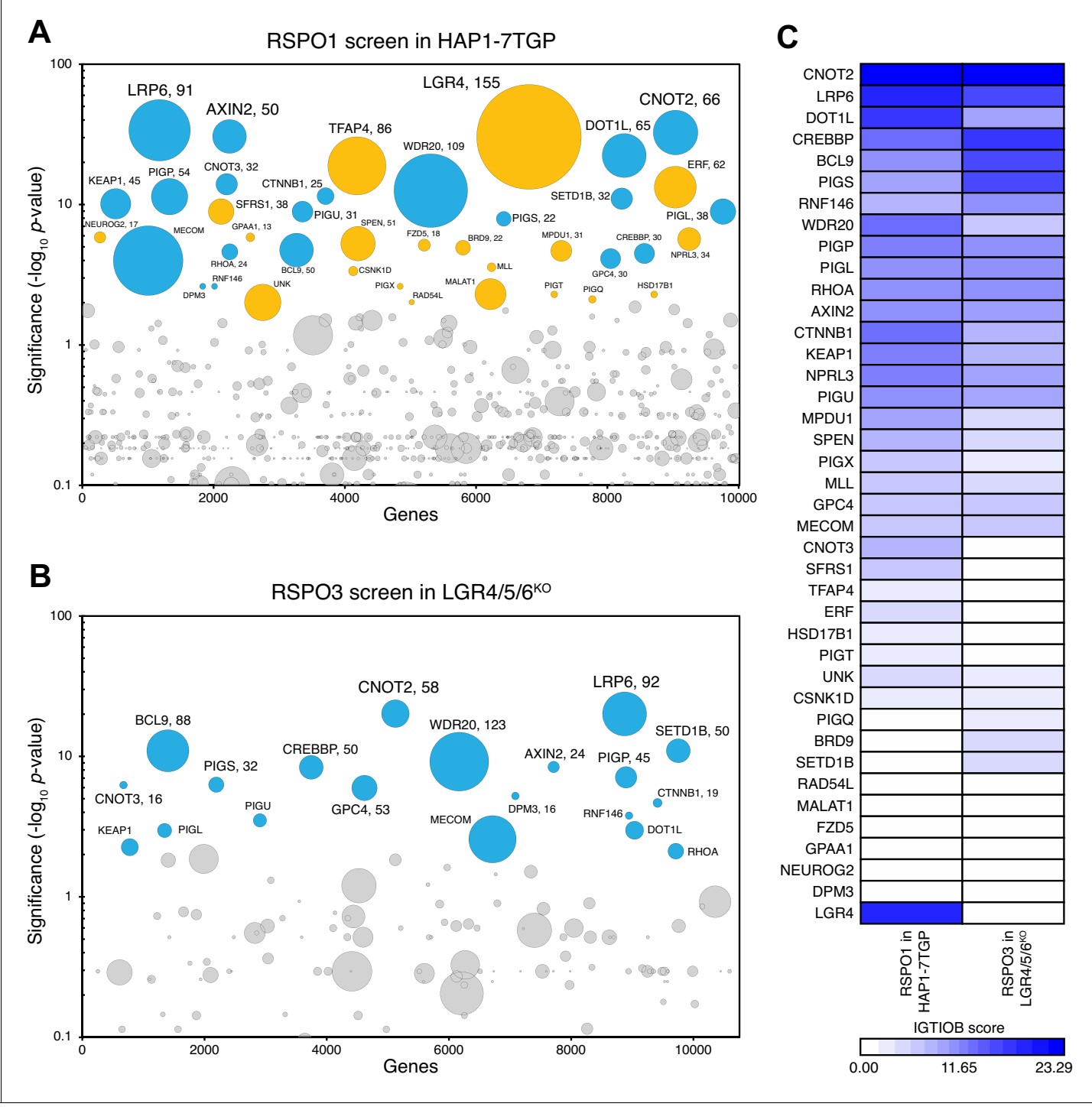

**Figure 6.** A haploid genetic screen for regulators of RSPO3 signaling in cells lacking LGRs. (A–B) Circle plots depicting genes enriched for inactivating gene trap (GT) insertions in the screen for regulators of RSPO1-potentiated WNT signaling in HAP1-7TGP cells (A) and in the screen for regulators of RSPO3-potentiated WNT signaling in LGR4/5/6$^{KO}$ cells (B). The y-axis indicates the significance of inactivating GT insertion enrichment in the sorted vs. the control cells (expressed in units of -log$_{10}$FDR-corrected *p*-value) and the x-axis indicates genes (in random order) for which inactivating GT insertions were mapped in the sorted cells (see Materials and methods). Genes with FDR-corrected *p*-value<0.01 are labeled and colored cyan if they reached this level of significance in both screens, or orange if they reached this level of significance in only one of the two screens. The diameter of each circle is proportional to the number of unique inactivating GT insertions mapped in the sorted cells, which is also indicated next to the gene name for the most significant hits with FDR-corrected *p*-values<10$^{-4}$. See **Supplementary file 4** for ranked lists of hits from both screens. (C) Heat map comparing the two

*Figure 6 continued on next page*

*Figure 6 continued*

screens. Genes enriched for inactivating GT insertions (FDR-corrected *p*-value<0.01) in at least one of the two screens (**Supplementary file 5**) were clustered based on their IGTIOB score in each screen (see Materials and methods).

## Discussion

The TSP/BR domain of RSPOs is conserved across vertebrates, primitive chordates and hemichordates, suggesting a functional role in signaling by these ligands (*de Lau et al., 2012*). The TSP/BR domain can bind to HS chains, but its role in amplifying WNT/β-catenin signaling has remained uncertain. Initial studies showed that this domain was dispensable, since a fragment of RSPOs that contains only the FU1 and FU2 domains can amplify WNT/β-catenin signaling in vitro (*Glinka et al., 2011*; *Kazanskaya et al., 2004*; *Kim et al., 2008*; *Zebisch et al., 2013*). Our previous (*Lebensohn and Rohatgi, 2018*) and current work provides a clear function for the TSP/BR domain of RSPO3 in WNT/β-catenin signaling in vitro and ex vivo: (1) it is essential for signaling in the absence of LGRs and (2) it enhances signaling potency in the presence of LGRs in both cultured cells and intestinal organoids. In this report we focused on RSPO3, but we previously demonstrated that RSPO2 (but not RSPO1 or RSPO4) can also amplify WNT/β-catenin signaling in the absence of LGRs. It will be interesting to determine whether the TSP/BR domains of RSPO1 and RSPO4 can also enhance ligand potency, especially in light of the observation that only RSPO2 and RSPO3, but not RSPO1 or RSPO4, bind to SDC1-4 (*Ohkawara et al., 2011*). Perhaps the divergent biological functions of different RSPOs (*de Lau et al., 2012*) is dictated by their ability to signal though HSPGs, LGRs or both. Importantly, we do not know how the RSPO concentrations in tissues compare to those typically used in in vitro culture systems, which can be arbitrarily high. Thus, the TSP/BR domain may be critical for certain RSPOs to potentiate WNT signaling in vivo, especially under regimes of limiting ligand concentration in tissues.

Until this study, it has been challenging to disentangle the role of HSPGs in mediating signaling through the core WNT/β-catenin pathway from their contribution to the amplification of WNT signaling by RSPOs. HSPGs and HS chains were initially implicated in the distribution, stability and potency of WNT ligands in studies from *Drosophila*, which lacks an RSPO signaling module for WNT amplification (*Baeg et al., 2001*; *Reichsman et al., 1996*). In those studies, HSPGs promoted WNT/β-catenin signaling by directly binding to WNT ligands. However, depletion of HS chains by the genetic disruption of enzymes required for their synthesis has also been shown to impair WNT/β-catenin signaling and associated responses in the mouse intestine, the mouse mammary gland and in multiple myeloma cells, all contexts in which RSPOs play prominent roles (*Alexander et al., 2000*; *Ren et al., 2018*; *Yamamoto et al., 2013*). Thus, a reduction in RSPO-potentiated WNT signaling caused by depletion of HSPGs can be due to a defect in WNT reception, RSPO reception or both. In fact, we showed previously (*Lebensohn et al., 2016*; *Lebensohn and Rohatgi, 2018*) that impaired WNT reception is the reason why genes encoding proteins required for the biogenesis of GPCs, and GPC4, the most abundant GPC in HAP1 cells, were prominent hits in both of our RSPO screens (*Figure 6A–C* and *Supplementary files 4* and *5*). Our present RSPO mutagenesis and engineering experiments, which leave the HSPG repertoire of cells intact and their role in WNT reception undisturbed, cleanly demonstrate that interactions with HSPGs are also required for RSPO reception.

How do HSPGs function during reception of RSPO ligands? HSPGs have been implicated in the function of many ligands (reviewed in *Sarrazin et al., 2011*), including chemokines and growth factors that activate receptor tyrosine kinases, such as fibroblast growth factor (FGF). In these cases, HSPGs can function as co-receptors by either templating a ligand-receptor interaction (such as the one between FGF and its receptor) or by inducing a conformational change in the ligand that increases its ability to interact with receptors (such as the one induced in anti-thrombin). In both of these cases, exogenously added heparin (untethered to the cell surface, in contrast to the HS chains of HSPGs) would be predicted to promote receptor-ligand interactions. Alternatively, HSPGs can function as endocytic receptors to mediate ligand uptake and degradation in the lysosome (*Belting, 2003*; *Sarrazin et al., 2011*; *Williams and Fuki, 1997*). Our results are more consistent with HSPGs functioning as endocytic receptors in RSPO signaling. First, exogenously added heparin inhibits, rather than promotes, RSPO signaling (*Lebensohn and Rohatgi, 2018*). Second, the TSP/BR domain can be replaced by a non-native, HS-binding scFv, making steric effects or conformational

changes unlikely. Instead, the mere interaction between RSPOs and HSPGs seems to be sufficient for signaling. Our results directly demonstrate that both WT RSPO3 and RSPO3 ΔTSP/BR HS20 can cause internalization and degradation of RNF43-Flag in cells lacking LGRs (*Figure 3B*). Since ZNRF3 and RNF43 are E3 ubiquitin ligases that negatively regulate WNT receptors, their clearance from the cell surface induced by RSPO3-HSPG interactions would be sufficient to enhance WNT signaling.

Our work also sheds light on the relative roles of HSPGs and LGRs in mediating signaling by different RSPO family members. Previous studies have proposed that the simultaneous binding of RSPOs to LGRs through the FU2 domain and to ZNRF3/RNF43 through the FU1 domain triggers the ubiquitination, endocytosis and degradation of ZNRF3/RNF43 (*Hao et al., 2012*; *Koo et al., 2012*). The affinity of the FU2 domain for LGRs is high (in the nM range) for all RSPOs, while the affinity of the FU1 domain for ZNRF3/RNF43 is much higher for RSPO2 and RSPO3 compared to RSPO1 and RSPO4 (*Zebisch et al., 2013*). At least for RSPO3, our experiments suggest that the interaction with LGRs is not sufficient to drive the efficient endocytosis of RNF43. HSPG binding, provided by the native TSP/BR domain or by the non-native HS20 scFv, is also required even in the presence of LGRs (*Figure 3A*). In the case of RSPO3 and also RSPO2, not only are LGRs insufficient to mediate RSPO signaling, but they are in fact dispensable. This is most likely due to the higher affinity of these RSPO ligands for ZNRF3/RNF43, since replacing the ZNRF3/RNF43-bindnig FU1 domain of RSPO1 with that of RSPO3 is sufficient to render RSPO1 capable of signaling without LGRs (*Lebensohn and Rohatgi, 2018*). However, even when LGRs are present, HSPGs can markedly enhance the potency of RSPO3 (*Figure 1D* and *Figure 5A and B*) and potentially other RSPOs. One possibility is that HSPGs trap RSPOs near the cell surface, increasing their local concentration and the extent of their binding to LGRs. This model is supported by the observation that either the genetic or enzymatic depletion of HS chains, or the removal of the TSP/BR domain, reduces binding of RSPOs to the surface of multiple myeloma cells, while the genetic depletion of LGR4 does not (*Ren et al., 2018*). Alternatively, the simultaneous binding of RSPO3 to LGRs and HSPGs may be required to trigger efficient endocytosis.

Several questions remain. We did not identify a single HSPG that serves as the RSPO receptor in HAP1 cells, most likely because HSPGs are multiply redundant in these cells (*Lebensohn and Rohatgi, 2018*). Disruption of GPC3, GPC4, all GPI-anchored GPCs or all SDCs either had no effect or only partially reduced the signaling potency of WT RSPO3 and synthetic RSPO3 ΔTSP/BR HS20 ligands (*Figure 4B and C*). Only the disruption of all HS-containing HSPGs reduced signaling by these RSPOs substantially, confirming the functional redundancy of HSPGs. However, specific HSPGs may be important in different tissues, stages of development or pathological conditions. For instance, the interaction of SDC4 and RSPO3 was shown to regulate WNT/planar cell polarity (PCP) signaling during *Xenopus* gastrulation by a process that requires clathrin-mediated endocytosis (*Ohkawara et al., 2011*), and SDC1 was shown to promote signaling in multiple myeloma by presenting WNTs and RSPOs (*Ren et al., 2018*). It will be interesting to test whether specific HSPGs mediate RSPO signaling during development of tissues that do not require the function of LGRs, such as the limbs and lungs (*Szenker-Ravi et al., 2018*). We also cannot discount the possibility that there are other undiscovered co-receptors capable of mediating the potentiation of WNT signaling by RSPOs during development. Our haploid genetic screens may have failed to identify an alternative RSPO co-receptor if it was redundant, required for cell viability or growth, or simply not expressed in HAP1 cells. These are known caveats of loss-of-function genetic screens.

Notwithstanding these possibilities, our current and previous results (*Lebensohn and Rohatgi, 2018*) strongly suggest that HSPGs such as GPCs or SDCs are the main co-receptors, in addition to ZNRF3 or RNF43, that transduce RSPO signals in the presence and absence of LGRs.

## Materials and methods

The following materials and methods relevant to this manuscript have been described previously (*Lebensohn et al., 2016*; *Lebensohn and Rohatgi, 2018*): cell lines and growth conditions, preparation of WNT3A CM, construction of the HAP1-7TGP WNT reporter haploid cell line, construction of mutant HAP1-7TGP cell lines by CRISPR/Cas9-mediated genome editing, production and immuno-affinity purification of tagged RSPO proteins by transient transfection of 293T cells, quantification of tagged RSPO proteins, reporter-based forward genetic screens and comparative analysis of

significant hits across screens. Where applicable, modifications to those materials and methods are described below.

## Key resources table

Provided as *Supplementary file 6*.

## Cell lines

Human haploid cells (HAP1) were validated using prodium iodide staining to ensure they have a haploid DNA content. All derivatives of HAP1 cells made using CRISPR/Cas9 methods were verified by DNA sequencing. HEK 293T cells were obtained from ATCC with a certificate of authentication and used at low passage (<10) without any additional STR profiling. Cell lines were confirmed to be mycoplasma-negative when cultures were started in the lab.

## Plasmids

pHLsec-HA-hRSPO3-Tev-Fc-Avi-1D4 and pHLsec-HA-hRSPO3ΔTSP/BR-Tev-Fc-Avi-1D4 (encoding proteins containing an N-terminal HA tag and C-terminal Fc and 1D4 tags) were constructed as described previously (*Lebensohn and Rohatgi, 2018*).

pHLsec-HA-hRSPO3TSP/BR(K/R→E)-Tev-Fc-Avi-1D4 (*Figure 1B* and *Supplementary file 1*) was synthesized as a gBlock Gene Fragment (Integrated DNA Technologies (IDT)) flanked at the 5' and 3' ends, respectively, by 24 base pair (bp) overhangs overlapping the sequence upstream of the unique AgeI site and downstream of the unique KpnI site in the pHLsec-HA-Tev-Fc-Avi-1D4 vector. The gBlock was subcloned into pHLsec-HA-Tev-Fc-Avi-1D4, previously digested with AgeI and KpnI, using the NEBuilder HiFi DNA Assembly Master Mix (New England Biolabs (NEB) Cat. # E2621L).

pHLsec-HA-hRSPO3ΔTSP/BRHS20-Avi-1D4, pHLsec-HA-hRSPO3ΔTSP/BRHS20(GS)-Avi-1D4, and pHLsec-HA-hRSPO3ΔTSP/BRHS20(A)-Avi-1D4, all of which encode proteins containing an N-terminal HA tag and a C-terminal 1D4 tag but lacking the dimerizing Fc tag (*Figure 2A* and *Supplementary file 1*), were synthesized as gBlock Gene Fragments (IDT) flanked at the 5' and 3' ends, respectively, by 24 bp overhangs overlapping the sequence upstream of the unique AgeI site and downstream of the unique KpnI site in the pHLsec-HA-Avi-1D4 vector described previously (*Lebensohn and Rohatgi, 2018*). The gBlock was subcloned into pHLsec-HA-Avi-1D4, previously digested with AgeI and KpnI, using the NEBuilder HiFi DNA Assembly Master Mix (NEB Cat. # E2621L).

pHLsec-HA-hRSPO3ΔTSP/BRHS20(R67A/Q72A)-Avi-1D4 and pHLsec-HA-hRSPO3ΔTSP/BRHS20 (F106E/F110E)-Avi-1D4 were made by replacing the FU1 and FU2 domains in pHLsec-HA-hRSPO3ΔTSP/BRHS20-Avi-1D4 with those from pHLsec-HA-hRSPO3(R67A/Q72A)-Tev-Fc-Avi-1D4 and pHLsec-HA-hRSPO3(F106E/F110E)-Tev-Fc-Avi-1D4 (*Lebensohn and Rohatgi, 2018*), respectively. A fragment containing the FU1 and FU2 domains was subcloned by double digestion with AgeI and MfeI followed by ligation.

RNF43-2xFlag-2xHA and N-terminal SNAP-tagged mouse FZD5 (Ala27-Val585) were described previously (*Koo et al., 2012*).

pVRC8400-GPC3ΔHS-hFc (pMH137) and pVRC8400-GPC4ΔHS-hFc (pMH373) were constructed as described previously (*Feng et al., 2013*). Briefly, serine to alanine substitutions were introduced at the HS attachment sites of GPC3 (S495A and S509A) or GPC4 (S494A, S498A and S500A) and the mutant constructs were PCR-amplified and inserted into the pVRC8400 expression vector (a gift from Dr. Gary J. Nabel, National Institute of Allergy and Infectious Diseases, Bethesda, MD). Sequences encoding the IL-2 signal peptide and a human Fc tag were added to the N-terminus and C-terminus of the mutant GPCs, respectively.

pX458-mCherry was made by replacing the coding sequence of GFP in pSpCas9(BB)−2A-GFP (pX458, a gift from Dr. Feng Zhang, Addgene plasmid # 48138) with that of a modified mCherry sequence. The WT mCherry sequence has a BbsI cleavage site that makes it incompatible with the BbsI restriction digestion required for cloning the single guide RNA (sgRNA) oligos into pX458. Therefore, a silent mutation was introduced into the mCherry sequence to eliminate the BbsI cleavage site. Terminal EcoRI sites and a T2A sequence were appended to the modified mCherry sequence to obtain an EcoRI-T2A-mCherry-EcoRI fragment. The EcoRI-T2A-EGFP-EcoRI fragment in

px458 was replaced with the EcoRI-T2A-mCherry-EcoRI fragment by EcoRI double digestion and ligation to obtain pX458-mCherry.

## Antibodies

Primary antibodies: Rho 1D4 purified monoclonal antibody (University of British Columbia, https://uilo.ubc.ca/rho-1d4-antibody), mouse anti-Flag M2 (MilliporeSigma Cat. # F3165), mouse anti-actin monoclonal (Clone C4, MP Biomedicals Cat. # 08691002), polyclonal rabbit anti-human lysozyme (Agilent Dako Cat. # A0099), rabbit anti-Ki67 (Abcam Cat. # ab15580).

Secondary antibodies: chicken anti-rabbit Alexa Fluor 488 (Thermo Fisher Scientific Cat. # A21441), goat anti-mouse Alexa Fluor 680 (Thermo Fisher Scientific Cat. # A21057), goat anti-mouse IRDye800 (Rockland Immunochemicals Cat. # 610-132-121), goat anti-human Ig kappa chain HRP conjugate (MilliporeSigma Cat. # AP502P).

Staining reagents: phalloidin-TRITC (MilliporeSigma Cat. # P1951), DAPI (MilliporeSigma Cat. # D9542).

## RSPO3 modeling

A structural model of human RSPO3 (residues valine 146 - isoleucine 232; UniProtKB Q9BXY4) was built using the HHpred web server (Max Planck Institute Bioinformatics Toolkit) based on homology to the 26 most similar protein structures in the Protein Data Bank (*Zimmermann et al., 2018*). Polymers consisting of four disaccharide units of heparin (PDB ID 1HPN [*Mulloy et al., 1993*]) were docked to RSPO3 using AutoDock Vina (*Trott and Olson, 2010*). Heparin was kept rigid during docking. Positively charged side chains of residues 161, 162, 164, 169, 170, 198, 203–205, 208, 211 (Site-1) or 213, 214, 216, 218–222, 225, 229 (Site-2) were kept flexible during docking. The models depicted in *Figure 1A* and the animation shown as *Figure 1—video 1* were rendered using PyMOL 2.3.2.

## Production of tagged RSPO proteins

Tagged RSPO proteins were produced by transient transfection of 293T cells and immuno-affinity purification as described previously (*Lebensohn and Rohatgi, 2018*), with some modifications. The WT, mutant and truncated RSPO3 proteins have an N-terminal HA tag and C-terminal Fc and 1D4 tags; the engineered RSPO3 ΔTSP/BR HS20 fusion proteins, and mutant derivatives thereof, have an N-terminal HA tag and a C-terminal 1D4 tag (see *Supplementary file 1* for nucleotide sequences). ~24 hr before transfection, $20 \times 10^6$ 293T cells were seeded in a T-225 flask containing 55 ml of growth medium (Dulbecco's Modified Eagle's Medium (DMEM)/high glucose without L-glutamine, sodium pyruvate (HyClone, GE Healthcare Life Sciences Cat. # SH30081.FS); 10% fetal bovine serum (FBS, MilliporeSigma Cat. # S11150); 1 mM sodium pyruvate (Gibco, Thermo Fisher Scientific Cat. # 11360070); 1x MEM non-essential amino acids (Gibco, Thermo Fisher Scientific Cat. # 11140050); 2 mM L-glutamine (Gemini Bio Products Cat. # 400–106)). Once they had reached ~80% confluency, the cells in each flask were transfected with 1 ml of a transfection mix prepared as follows. 30 μg of DNA constructs encoding tagged WT, mutant or synthetic RSPO proteins were diluted in 910 μl of serum-free DMEM and 90 μl of polyethylenimine (PEI, linear, MW ~25,000, Polysciences, Inc Cat. # 23966) was added from a 1 mg/ml stock (prepared in sterile water, stored frozen and equilibrated to 37°C before use). The transfection mix was vortexed briefly, incubated for 15–20 min at room temperature (RT) and added to the cells without replacing the growth medium. ~24 hr post-transfection, the cells were washed with 55 ml PBS and the medium was replaced with 55 ml of CD 293 medium (Thermo Fisher Scientific Cat. # 11913019) supplemented with 1x L-glutamine solution and 2 mM valproic acid (MilliporeSigma Cat. # P4543, added from a 0.5 M stock prepared in water and sterilized by filtration through a 0.22 μm filter) to promote protein expression. ~96 hr post-transfection, the CM from each flask, containing secreted RSPO protein, was filtered through 0.45 μm filters (Nalgene aPES membrane, Thermo Fisher Scientific Cat. # 166–0045) to remove particulates and was reserved on ice. 300 μl of a ~ 50% slurry of Rho 1D4 resin (prepared as described in *Lebensohn and Rohatgi, 2018*) was added to a 50 ml conical tube containing the RSPO CM and the suspension was incubated overnight, rocking at 4°C. Following binding and during all subsequent washes, the resin was collected by centrifugation for 5 min at 400 x g in a swinging bucket rotor. The beads were washed three times at RT with 25 ml PBS by resuspending them in

buffer and mixing by inversion for ~1 min. The resin was transferred to a 1.5 ml Eppendorf tube and washed three more times with 1 ml of PBS, 10% glycerol. Following the last wash, the buffer was aspirated and the resin was resuspended in 150 µl of 500 µM 1D4 peptide ((NH3)-T-E-T-S-Q-V-A-P-A-(COOH)) prepared in PBS, 10% glycerol, to obtain a ~ 50% slurry. Elution was carried out by rotating the tube horizontally overnight at 4℃. Following centrifugation of the resin, the eluate was recovered and reserved on ice. The resin was resuspended in another 150 µl of 500 µM 1D4 peptide in PBS, 10% glycerol, and a second round of elution was carried out for 2 hr at RT. Following centrifugation of the resin, the second eluate was recovered and pooled with the first. The final eluate was centrifuged again to remove residual resin and the supernatant containing tagged RSPO proteins was aliquoted, frozen in liquid nitrogen and stored at −80℃. Tagged RSPO proteins were quantified as described previously (*Lebensohn and Rohatgi, 2018*).

## Dose-response analysis of RSPO3 effects on WNT reporter fluorescence

20,000 cells were plated per well of a 96-well plate and grown in complete IMDM growth medium (Iscove's Modified Dulbecco's Medium (IMDM) modified with L-glutamine, HEPES (GE Healthcare Life Sciences Cat. # SH30228.01); 2 mM L-glutamine; 40 units/ml penicillin:40 µg/ml Streptomycin (Gemini Bio-products Cat. # 400–109); 10% FBS). ~24 hr after plating, the cells were treated with complete IMDM growth medium containing 1.43% WNT3A CM and the indicated concentrations of RSPO proteins. After 24 hr of treatment, cells were trypsinized in 40 µl of Tryspin-EDTA (0.05%, Thermo Fisher Scientific Cat. # 25300054) and harvested in 160 µl of complete IMDM growth medium. WNT reporter (EGFP) fluorescence data for 2,500 singlet-gated cells was measured by flow cytometry on a BD Accuri RUO Special Order System (BD Biosciences). The median EGFP fluorescence from each well was depicted as circles in *Figure 1D and E* and *Figure 2C–H*. Dose-response curves were fitted using the nonlinear regression (curve fit) analysis tool in GraphPad Prism 8 using the [agonist] vs. response – variable slope (four parameters) equation. Results presented are representative of experiments repeated at least twice.

## Cell-surface RNF43 biotinylation assay

WT HAP1-7TGP or LGR4/5/6$^{KO}$ cells were transfected with human RNF43-2xFlag-2xHA and SNAP-tagged mouse FZD5 using PEI (*Koo et al., 2012*). 24 hr after transfection, cells were washed with ice-cold PBS complete (containing Ca$^{2+}$ and Mg$^{2+}$) and kept on ice. 800 µM of EZ-Link Sulfo-NHS-SS-Biotin (Thermo Fisher Scientific Cat. # 21331) in PBS complete was added to each well and biotinylation was allowed to proceed for 30 min on ice. The medium was aspirated, cells were washed once with ice-cold PBS complete and unreacted biotin was quenched with complete IMDM growth medium containing 50 mM glycine for 5 min on ice. Cells were washed with ice-cold PBS complete and lysed for 30 min on ice in lysis buffer (20 mM Tris pH 7.5, 150 mM NaCl, 1% Triton X-100, 1 mM EDTA, 1 mM PMSF, 10 µg/ml Leupeptin, 10 µg/ml Aprotinin). Cell lysates were centrifuged for 5 min at 14,000 x g at 4℃. An aliquot of the supernatant ('Total' in *Figure 3A and B*) was mixed with SDS sample buffer and the remaining supernatant was incubated with 25 µl of a 50% slurry of Streptavidin agarose beads (Pierce, Thermo Fisher Scientific Cat. # 20349) for 60 min at 4℃. The beads were washed with 0.1x PBS complete, and biotinylated proteins ('Surface' in *Figure 3A and B*) were eluted in 20 µl 2x SDS sample buffer, 40 mM DTT for 30 min at 37℃ and analyzed by SDS-PAGE and Western blot. Western blotting was performed using standard procedures. Samples were resolved by SDS-PAGE, transferred to Immobilon-FL PVDF membranes (MiliporeSigma Cat. # IPFL00005), blocked with Odyssey blocking buffer (Li-Cor Cat. # 927–40000), incubated with the indicated primary and secondary antibodies, and imaged using the Amersham Typhoon NIR laser scanner (GE Healthcare).

## Production of recombinant GPC mutant proteins without HS chains

The GPC3 and GPC4 mutant proteins without HS chains (GPC3ΔHS and GPC4ΔHS in *Figure 4A*) were constructed by replacing serine with alanine residues at the HS attachment sites. pVRC8400-GPC3ΔHS-hFc or pVRC8400-GPC4ΔHS-hFc were transiently transfected into HEK-293T cells. The cell supernatant was harvested and the Fc-tagged GPC proteins were purified on a Protein A Hi-Trap column (GE Healthcare Cat. # 29048576) according to the manufacturer's instructions. The

quality and quantity of purified proteins were determined by SDS-PAGE and by measuring absorbance at 280 nm on a NanoDrop instrument (Thermo Fisher Scientific), respectively.

## Measurement of HS20 binding to GPCs by enzyme-linked immunosorbent assay (ELISA)

Increasing concentrations of HS20 (serial 2-fold dilutions starting from 1 µg/ml) were added into a 96-well ELISA plate coated with 5 µg/ml of glycosylated GPC3, GPC4 (R and D Systems Cat. # 2119-GP and 9195-GP, respectively), GPC3ΔHS, GPC4ΔHS, or an irrelevant human Fc-fusion protein (CD276-hFc, labeled 'Control' in *Figure 4A*) and incubated for 1 hr at RT. The plate was washed three times with PBST (PBS, 0.1% Tween-20) and incubated with 50 µl of goat anti-human kappa chain HRP conjugate (1:5,000 dilution) for 1 hr at RT. After washing six times with PBST, 50 µl/well of 3,3′,5,5′-tetramethylbenzidine detection reagent (Kirkegaard and Perry Laboratories Cat. # 95059–156) was added and incubated for 10 min at RT. Absorbance was read at 450 nm.

## Construction of mutant HAP1-7TGP cell lines by CRISPR/Cas9-mediated genome editing

The LGR4/5/6$^{KO}$, LGR4/5/6$^{KO}$; PIGL$^{KO}$, LGR4/5/6$^{KO}$; SDC1/2/3/4$^{KO}$ and LGR4/5/6$^{KO}$; EXTL3$^{KO}$ cell lines used in this study have been described previously (*Lebensohn and Rohatgi, 2018*). The LGR4/5/6$^{KO}$; GPC3$^{KO}$ and LGR4/5/6$^{KO}$; GPC4$^{KO}$ cell lines were constructed as described previously (*Lebensohn et al., 2016*). Briefly, oligonucleotides encoding sgRNAs (*Supplementary file 2*) were selected from a published library (*Doench et al., 2016*) and cloned into pX458-mCherry according to a published protocol (*Cong et al., 2013*) (original version of 'Target Sequence Cloning Protocol' from http://www.genome-engineering.org/crispr/wp-content/uploads/2014/05/CRISPR-Reagent-Description-Rev20140509.pdf). Clonal cell lines were established as described previously (*Lebensohn et al., 2016*) by transient transfection of LGR4/5/6$^{KO}$ cells with pX458-mCherry containing the desired sgRNAs (since we used pX458-mCherry, there was no need to co-transfect with pmCherry), followed by single-cell sorting of mCherry-positive cells. Genotyping was done as described previously (*Lebensohn et al., 2016*) using the primers indicated in *Supplementary file 2*, and the sequencing results are summarized in *Supplementary file 3*.

## Growth of mouse small intestinal organoids

Small intestinal organoids were established and maintained as described (*Sato et al., 2009*), starting from isolated crypts collected from the entire length of the small intestine of B6 mice. The basic culture medium (advanced DMEM/F12, penicillin/streptomycin, 10 mM HEPES, 1x GlutaMAX, 1x N-2 supplement, 1x B-27 supplement (all from Gibco, Thermo Fisher Scientific Cat. # 12634010, 400–109, 15630080, 35050061, 17502001 and 17504044, respectively), 1 mM N-acetylcysteine (Millipore-Sigma Cat. # 106425)), was supplemented with 50 ng/ml recombinant murine EGF (Peprotech Cat. # 315–09), and 1% v/v Noggin CM to obtain EN medium. Purified RSPO recombinant proteins were added to the basic culture medium as indicated. Organoids were cultured in matrigel droplets (Corning CellBIND, Corning Cat. # 3300, 3335, 3292). The medium was refreshed 3 and 6 days after splitting. Images were captured 8 days after splitting using an EVOS M5000 imaging system (Thermo Fisher Scientific).

## Immunofluorescence microscopy of organoids

Organoids were carefully harvested from matrigel, collected by centrifugation for 5 min at 100 x g at 4°C and washed with ice-cold medium. Organoids were transferred to a µ-slide 8 well (Ibidi Cat. # 80826) and fixed in paraformaldehyde (4%, diluted in 0.1 M sodium phosphate buffer pH 7.4) for 1 hr at RT. Paraformaldehyde was removed and 20 mM ammonium chloride (in PBS) was added for 10 min. Organoids were permeabilized in PBD 0.2 T buffer (1x PBS, 1% BSA, 1% DMSO, 0.2% TX-100) for 30 min at RT. Organoids were incubated with anti-human lysozyme or anti-Ki67 primary antibodies in PBD 0.2 T for 3 hr at RT, followed by anti-rabbit Alexa Fluor 488 secondary antibody containing 0.2 µg/ml DAPI and phalloidin-TRITC in PBD 0.2 T for 3 hr at RT. Organoids were mounted in Ibidi mounting medium (Ibidi Cat. # 50001) and images were acquired with a Zeiss LSM700 confocal microscope. Images were processed in ImageJ (National Institutes of Health).

# Reporter-based haploid genetic screens for regulators of RSPO1 signaling in HAP1-7TGP cells and for regulators of RSPO3 signaling in LGR4/5/6^KO cells

HAP1-7TGP or LGR4/5/6$^{KO}$ cells were mutagenized with a GT-containing retrovirus as described previously (*Lebensohn et al., 2016*). Six days following mutagenesis, the HAP1-7TGP or LGR4/5/6$^{KO}$ cell populations were treated with a low dose (1.04%) of WNT3A CM in complete IMDM growth medium, combined with a saturating dose (10 ng/ml) of recombinant human RSPO1 (R and D Systems Cat. # 4645-RS) or RSPO3 (R and D Systems Cat. # 3500-RS), respectively. A total of $1.25 \times 10^8$ singlet-gated cells were sorted by fluorescence-activated cell sorting (FACS), gating for the lowest ~10% WNT reporter (EGFP) fluorescence. A population of singlet-gated HAP1-7TGP or LGR4/5/6$^{KO}$ cells (not gated based on EGFP fluorescence) was also sorted and carried along as a control to set FACS gates during the subsequent sort. Cells were expanded for 6 days to allow recovery and resetting of the WNT reporter, and a portion of the cells was used in a subsequent round of sorting, following the same treatment and FACS gating criteria. For both screens described, cells underwent two consecutive rounds of FACS sorting. $3 \times 10^7$ cells expanded following the final sort for each screen were used to identify GT insertions in the sorted cell populations and an equivalent number of mutagenized but unsorted HAP1-7TGP or LGR4/5/6$^{KO}$ cells were used to identify GT insertions in the respective control cell populations. Preparation of genomic GT insertion libraries for deep sequencing, mapping of sequencing reads and statistical analysis of GT insertion enrichment was conducted as described previously (*Lebensohn et al., 2016*), except that the libraries for the unsorted (control) and sorted cell populations from the RSPO3 screen in LGR4/5/6$^{KO}$ cells were indexed and sequenced together in an Illumina NextSeq 500 system. FASTQ files containing sequencing data for the unsorted and sorted cell pupulations from the RSPO3 screen in LGR4/5/6$^{KO}$ cells, as well as for the sorted cell population from the RSPO1 screen in HAP1-7TGP cells have been deposited in the National Center for Biotechnology Information (NCBI) Sequence Read Archive (SRA) with study accession number SRP260445. The FASTQ file containing the sequencing data for the unsorted cell pupulation from the RSPO1 screen in HAP1-7TGP cells had been deposited previously with study accession number SRP094861 (SRA accession number SRX2410629). We generated a ranked list of hits from each screen (*Supplementary file 4*) and used it to create circle plots depicting the data (*Figure 6A and B*). Note that the data presented here for the RSPO1 screen in HAP1-7TGP cells is different from that presented in our previous report (*Lebensohn et al., 2016*) because in order to maintain a comparable degree of phenotypic enrichment between this screen and the RSPO3 screen in LGR4/5/6$^{KO}$ cells, we analyzed a cell population from our previous RSPO1 screen in HAP1-7TGP cells that had been sorted only twice (as opposed to four times) for the lowest ~10% WNT reporter activity.

## Comparative analysis of significant hits between screens

We used two metrics to compare the results of the RSPO1 screen in HAP1-7TGP cells and the RSPO3 screen in LGR4/5/6$^{KO}$ cells. The FDR-corrected *p*-value (*Figure 6A and B*, and *Supplementary file 4*) tests for the enrichment of inactivating GT insertions in the sorted cell population over total GT insertions in the control cell population (*Lebensohn et al., 2016*). While this is a good metric by which to compare the significance of hits in a given screen, it depends on many experimental variables affecting the identification of GT insertions in both the sorted and control cell populations, making it an equivocal metric to directly compare hits between screens. We therefore used the Intronic GT Insertion Orientation Bias (IGTIOB) score we developed previously (*Lebensohn et al., 2016*), which is based on the frequency of intronic sense vs. antisense GT insertions only in the sorted cell population from each screen, to directly compare significant hits between the RSPO1 screen in HAP1-7TGP cells and the RSPO3 screen in LGR4/5/6$^{KO}$ cells. In order to avoid missing any potential hits unique to only one screen, we relaxed the GT enrichment FDR-corrected *p*-value threshold for inclusion in the comparative analysis from $<10^{-4}$ (used in our previous studies [*Lebensohn et al., 2016*]) to <0.01. We generated a list of all genes with an FDR-corrected *p*-value<0.01 in at least one of the two screens (*Supplementary file 5*) and used the IGTIOB scores of those genes in both screens to build a heat map (*Figure 6C*) as described previously (*Lebensohn et al., 2016*).

## Statistical analysis

Data analysis and visualization for the dose-response analysis of RSPO3 effects on WNT reporter fluorescence (*Figure 1D and E*, *Figure 2C–H*) were performed in GraphPad Prism 8. Dose-response curves were fitted with the non-linear regression curve fit using the [agonist] vs. response – variable slope (four parameters) equation. All reported $EC_{50}$ values were obtained from these curve fits. The statistical significance of the differences in $EC_{50}$ values for various RSPO variants and cell lines (*Figure 4B and C*) was determined by a two-tailed, unpaired t-test in GraphPad Prism 8. Statistical analysis of screen data was performed as described previously (*Lebensohn et al., 2016*).

## Acknowledgements

RR was supported by grants from the National Institutes of Health (GM118082), AML and MH by the Intramural Research Program of the National Institutes of Health, National Cancer Institute, Center for Cancer Research, MMM by grants from the Netherlands Organization for Scientific Research (NWO VICI Grant 91815604 and TOP Grant 91218050), CS by grants from Cancer Research UK (C20724/A14414 and C20724/A26752) and a European Research Council grant (647278), and JEC by a grant from the National Institutes of Health (AI141970). The work of MMM, PvK and IJ is part of the Oncode Institute, which is partly financed by the Dutch Cancer Society. JEC is a Burroughs Wellcome Investigator in the pathogenesis of infectious disease. GVP was supported by a postdoctoral fellowship from the American Heart Association (14POST20370057) and RD by fellowships from the Stanford Dean's Fund and the Alex's Lemonade Stand Foundation.

## Additional information

### Funding

| Funder | Grant reference number | Author |
|---|---|---|
| National Institutes of Health | GM118082 | Ramin Dubey<br>Ganesh V Pusapati<br>Rajat Rohatgi |
| National Institutes of Health | Intramural Research Program | Mitchell Ho<br>Andres M Lebensohn |
| American Heart Association | 14POST20370057 | Ganesh V Pusapati |
| Alex's Lemonade Stand Foundation for Childhood Cancer | postdoctoral fellowship grant | Ramin Dubey |
| Cancer Research UK | C20724/A14414 | Christian Siebold |
| European Research Council | 647278 | Christian Siebold |
| National Institutes of Health | AI141970 | Jan E Carette |
| Burroughs Wellcome Fund | Investigator in the pathogenesis of infectious diseases | Jan E Carette |
| Dutch Cancer Society | Oncode Institute | Peter van Kerkhof<br>Ingrid Jordens<br>Madelon Maurice |
| Nederlandse Organisatie voor Wetenschappelijk Onderzoek | NWO VICI Grant 91815604 | Madelon Maurice |
| Cancer Research UK | C20724/A26752 | Christian Siebold |
| Nederlandse Organisatie voor Wetenschappelijk Onderzoek | TOP Grant 91218050 | Madelon Maurice |
| Stanford Dean's Fund | | Ramin Dubey |

The funders had no role in study design, data collection and interpretation, or the decision to submit the work for publication.

## Author contributions

Ramin Dubey, Conceptualization, Formal analysis, Validation, Investigation, Visualization, Methodology, Writing - review and editing; Peter van Kerkhof, Ingrid Jordens, Conceptualization, Formal analysis, Investigation, Visualization, Methodology, Writing - review and editing; Tomas Malinauskas, Christian Siebold, Formal analysis, Visualization, Methodology; Ganesh V Pusapati, Investigation, Methodology, Writing - review and editing; Joseph K McKenna, Resources, Investigation; Dan Li, Investigation, Visualization; Jan E Carette, Resources, Formal analysis, Methodology; Mitchell Ho, Resources, Methodology; Madelon Maurice, Conceptualization, Resources, Formal analysis, Writing - review and editing; Andres M Lebensohn, Conceptualization, Resources, Formal analysis, Supervision, Validation, Investigation, Visualization, Methodology, Writing - original draft, Writing - review and editing; Rajat Rohatgi, Conceptualization, Resources, Formal analysis, Supervision, Funding acquisition, Writing - original draft, Project administration, Writing - review and editing

## Author ORCIDs

Ramin Dubey (iD) https://orcid.org/0000-0002-0687-5831
Tomas Malinauskas (iD) http://orcid.org/0000-0002-4847-5529
Ganesh V Pusapati (iD) http://orcid.org/0000-0002-1406-2566
Joseph K McKenna (iD) https://orcid.org/0000-0002-3921-4933
Mitchell Ho (iD) https://orcid.org/0000-0002-9152-5405
Christian Siebold (iD) http://orcid.org/0000-0002-6635-3621
Andres M Lebensohn (iD) https://orcid.org/0000-0002-4224-8819
Rajat Rohatgi (iD) https://orcid.org/0000-0001-7609-8858

## Decision letter and Author response

Decision letter https://doi.org/10.7554/eLife.54469.sa1
Author response https://doi.org/10.7554/eLife.54469.sa2

# Additional files

## Supplementary files

• Supplementary file 1. Nucleotide sequences of RSPO3 WT, mutant and HS20-fusion constructs used in this study. The name of the encoded protein and the length (in bp) of the nucleotide sequence is indicated. RSPO3 (WT), RSPO3 TSP/BR (K/R→E) and RSPO3 ΔTSP/BR were cloned into pHLsec-HA-Tev-Fc-Avi-1D4. RSPO3 ΔTSP/BR HS20, RSPO3 ΔTSP/BR HS20 (GS), RSPO3 ΔTSP/BR HS20 (A), RSPO3 ΔTSP/BR HS20 (R67A/Q72A) and RSPO3 ΔTSP/BR HS20 (F106E/F110E) were cloned into pHLsec-HA-Avi-1D4. Bases in lowercase overlap the sequences upstream of the unique AgeI sites and downstream of the unique KpnI sites in the pHLsec-HA-Tev-Fc-Avi-1D4 and pHLsec-HA-Avi-1D4 vectors, respectively. Bases in uppercase encode RSPO3 WT, mutant and HS20-fusion proteins. For mutant constructs, mutated bases are indicated in red and the resulting altered codons are underlined. For HS20-fusion constructs, bases encoding a codon-optimized glycine/serine linker (STGGSGGSGGSG) are indicated in light blue.

• Supplementary file 2. List of oligonucleotides and primers used to generate and characterize clonal cell lines engineered using CRISPR/Cas9. The names and sequences of pairs of oligonucleotides encoding sgRNAs (which were cloned into pX458-mCherry) are shown in the first and second columns, respectively. The names and sequences of pairs of PCR primers used to amplify corresponding genomic regions flanking sgRNA target sites are shown in the third and fourth columns, respectively. The names and sequences of primers used to sequence the amplified target sites are shown in the fifth and sixth columns, respectively.

• Supplementary file 3. Description of engineered cell lines used in this study. Clonal cell lines derived from HAP1-7TGP in which multiple genes were targeted using CRISPR/Cas9 (see Materials and methods) are described. The 'Cell Line Name' column indicates the generic name used throughout the manuscript to describe the genotype and the 'Clone #' column identifies the individual clone used. The figures in which each clone was used are also indicated. The 'CRISPR guide' column indicates the name of the guide or guides used, which is the same as that of the oligonucleotides

encoding sgRNAs (see Materials and methods and *Supplementary file 2*). The 'Genomic Sequence' column shows 80 nucleotides of genomic sequence (5' relative to the gene is to the left) surrounding the target site; when two adjacent sites within the same gene were targeted, 80 nucleotides of genomic sequence surrounding each target site are shown and the number of intervening bp that are not shown between the two sites is indicated in parenthesis. Each cell line made using a different set of CRISPR guides is separated by a horizontal spacer, under which the reference (WT) genomic sequence (obtained from RefSeq) targeted by each CRISPR guide is indicated. Within this reference genomic sequence, the guide sequence is colored blue and the site of the double strand cut made by Cas9 is between the two underlined bases. Sequencing results for individual mutant clones are indicated below the reference sequence. Mutated, inserted or deleted nucleotides are colored red (dashes represent deleted nucleotides and ellipses are used to indicate that a deletion continues beyond the 80 nucleotides of sequence shown) and the nature of the mutation, the resulting genotype and any pertinent observations are also described. The CRISPR guide or guides used to target different genes, as well as the genomic sequence, mutation, genotype and observations pertaining to each of the targeted genes are designated '1', '2', '3' and '4' in the column headings and are shown under horizontal spacers of different colors.

• Supplementary file 4. Ranked lists of hits from screens. Genes containing at least one inactivating GT insertion in the population of sorted cells from each of the two genetic screens described in this work are listed in separate spreadsheets (the screen name is indicated on the tab of each spreadsheet), and are ranked based on the significance of inactivating GT insertion enrichment (*p*-value) in the sorted vs. the unsorted (control) cells. For the unsorted cells, the number of all GT insertions in genes (regardless of orientation) is indicated for the complete dataset and for each gene (column B). For the sorted cells, the total number of inactivating GT insertions in genes (sense and antisense insertions in exons and sense insertions in introns, column C), as well as the number of sense or antisense GT insertions in exons or in introns (columns D-G), is indicated for the complete dataset and for each gene. Three measures of GT insertion enrichment are shown: the *p*-value and the FDR-corrected *p*-value (both derived from columns B and C), the latter of which was used to generate the circle plots in *Figure 6A and B*, and the Intronic GT Insertion Orientation Bias (IGTIOB) score (derived from columns F and G), used to generate the heat map in *Figure 6C*. See Materials and methods for details.

• Supplementary file 5. List of significant hits included in the comparative analysis between screens. Genes used to generate the heat map in *Figure 6C*, comparing the RSPO1 screen in WT HAP1-7TGP cells (*Figure 6A*) and the RSPO3 screen in LGR4/5/6$^{KO}$ cells (*Figure 6B*), are shown. Genes enriched for inactivating GT insertions (FDR-corrected *p*-value<0.01) in at least one of the two screens are shown, and the FDR-corrected *p*-value and IGTIOB score for each gene in each screen is indicated. Genes are shown in the same order as in the heat map in *Figure 6C*, clustered based on their IGTIOB scores (see Materials and methods for details).

• Supplementary file 6. Key resources table.

• Transparent reporting form

## Data availability

All of the data generated and analysed in the manuscript has been provided in the manuscript itself and in the accompanying supplementary files. FASTQ files containing sequencing data from the haploid genetic screens (see Materials and methods) have been deposited in the National Center for Biotechnology Information (NCBI) Sequence Read Archive (SRA) with study accession numbers SRP260445 and SRP094861.

The following dataset was generated:

| Author(s) | Year | Dataset title | Dataset URL | Database and Identifier |
|-----------|------|---------------|-------------|--------------------------|
| Dubey R, van Kerkhof P, Jordens I, Malinauskas | 2020 | R-spondins engage heparan sulfate proteoglycans to potentiate WNT signaling | https://trace.ncbi.nlm.nih.gov/Traces/sra/?study=SRP260445 | NCBI Sequence Read Archive (SRA), SRP260445 |

T, Pusapati GV,
McKenna JK, Li D,
Carette JE, Ho M,
Siebold C, Maurice
M, Lebensohn AM,
Rohatgi R

The following previously published dataset was used:

| Author(s) | Year | Dataset title | Dataset URL | Database and Identifier |
|---|---|---|---|---|
| Lebensohn AM, Dubey R, Neitzel LR, Tacchelly-Benites O, Yang E, Marceau CD, Davis EM, Patel BB, Bahrami-Nejad Z, Travaglini KJ, Ahmed Y, Lee E, Carette JE, Rohatgi R | 2016 | Comparative genetic screens in human cells reveal new regulatory mechanisms in WNT signaling | https://trace.ncbi.nlm.nih.gov/Traces/sra/?study=SRP094861 | NCBI Sequence Read Archive (SRA), SRP094861 |

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
