## [Decision Letter]

**Acceptance summary:**

This manuscript from Dubey and colleagues presents a significant extension of their interesting observations from an *eLife* paper published in 2018. In the current work, they use a creative approach to show that R-spondin3 signaling that is independent of LGR4/5/6 requires interactions with Heparan sulfate proteoglycans (HSPGs). Overall, the work is well executed from a technical perspective, innovative approaches are utilized, and the manuscript is well written.

**Decision letter after peer review:**

Thank you for submitting your article "R-spondins engage heparan sulfate proteoglycans to potentiate WNT signaling" for consideration by *eLife*. Your article has been reviewed by two peer reviewers, and the evaluation has been overseen by a Reviewing Editor and Jonathan Cooper as the Senior Editor. The following individual involved in review of your submission has agreed to reveal their identity: Bart Williams (Reviewer #1).

The reviewers have discussed the reviews with one another and the Reviewing Editor has drafted this decision to help you prepare a revised submission.

Summary:

This manuscript from Dubey and colleagues presents information that extends their interesting observations from an e*Life* paper published in 2018. In this work, they use a creative approach to show that R-spondin3 signaling that is independent of LGR4/5/6 requires interactions with Heparan sulfate proteoglycans (HSPGs). Overall the results are of good quality and the findings are interesting and represent sufficient new progress.

Essential revisions:

1) Some additional information and controls related to HS20 and the rationale for choosing GPC3 as a target for the fusion protein would further enhance the manuscript. What was the rationale for specifically choosing GPC3 as the HSPG that was chosen to target in this work? What is the level of GPC3 expression in the systems (cell lines and organoids) chosen for this work? How does it compare to GPC4 (which the authors note was identified as a hit in both the RSPO1 and RSPO3 haploid genetic screens)?

2) The fusion of the mutant RSPO3 with HS20 is innovative. What is known about its specificity for GPC3 relative to other GPCs? Further evidence for specificity could be provided by knocking down GPC3 levels in the cell line and/or organoid systems and showing that the signaling activity was the RSPO3-ΔTSP/BR-HS20 fusion protein was significantly reduced or lost. Alternatively, the authors could carry out the haploid genetic screen with the RSPO3-ΔTSP/BR-HS20 fusion protein to see if GPC3 is now required for activity.

3) The authors speculate on an interesting point that the ability of the different R-spondins to signal is "dictated by their ability to signal through HSPGs, LGRs, or both." Is there evidence that RSPO3 can interact with HSPGs and LGRs simultaneously? Perhaps examining whether the RSPO3-ΔTSP/BR-HS30 fusion protein can simultaneously interact with LGR4/5/6 and its known target GPC3 could help address this?

4) Given the inherent difficulties of interpreting negative results, the negative results Figure 4 may be minimized and presented as a figure supplement.

---

## [Author Response]

Essential revisions:1) Some additional information and controls related to HS20 and the rationale for choosing GPC3 as a target for the fusion protein would further enhance the manuscript. What was the rationale for specifically choosing GPC3 as the HSPG that was chosen to target in this work? What is the level of GPC3 expression in the systems (cell lines and organoids) chosen for this work? How does it compare to GPC4 (which the authors note was identified as a hit in both the RSPO1 and RSPO3 haploid genetic screens)?

Though HS20 was originally selected for binding to GPC3, we have previously presented data showing that it recognizes the Heparan Sulfate (HS) chains attached to GPC3 (Gao et al., 2014). Thus, we chose HS20 because it was likely to bind to many different HSPGs (which have different core proteins but share HS chains of similar composition). In the revised manuscript, we thoroughly tested the HSPG specificity of HS20 using binding assays and a panel of cell lines carrying increasingly severe deficits in their HSPG composition. This new data are presented in Figure 4 (entirely new in the revision) and discussed in the new section of the Results entitled “HSPG specificity of RSPO3 signaling”. These data show that RSPO3 ΔTSP/BR HS20 does not bind to GPC3 specifically; instead, it can use multiple different HSPGs to potentiate WNT signaling, likely by binding to their HS chains (new Figures 4B and C). To directly address the specific GPC3 vs. GPC4 question raised above, we performed binding assays with purified proteins (new Figure 4A) to show that HS20 binds equally to GPC3 and GPC4, but not to mutant proteins lacking the HS chains.

We have now also included text describing expression levels of GPC3 and other HSPGs in both HAP1 cells and in mouse intestinal organoids, the two systems used in our study. HAP1 cells (as originally presented in Table 1 of our precursor *eLife* paper (Lebensohn and Rohatgi, 2018)), express both *GPC3* and *GPC4*, with *GPC4* mRNA being more abundant. In mouse intestinal organoids, a prior publication from a different group found that only *GPC4* (but not *GPC3*) is expressed, supporting the model that RSPO3 ΔTSP/BR HS20 is not specific for GPC3 (Lindeboom et al., 2018).

2) The fusion of the mutant RSPO3 with HS20 is innovative. What is known about its specificity for GPC3 relative to other GPCs? Further evidence for specificity could be provided by knocking down GPC3 levels in the cell line and/or organoid systems and showing that the signaling activity was the RSPO3-ΔTSP/BR-HS20 fusion protein was significantly reduced or lost. Alternatively, the authors could carry out the haploid genetic screen with the RSPO3-ΔTSP/BR-HS20 fusion protein to see if GPC3 is now required for activity.

Please see the response to Essential revision #1 above, which describes our characterization of the HSPG specificity of HS20 (data shown in a new Figure 4 and discussed in the section entitled “HSPG specificity of RSPO3 signaling”). We knocked out GPC3 in LGR4/5/6^KO^ cells (Figures 4B and C) and found no effect on signaling by RSPO3 ΔTSP/BR HS20. Also, GPC3 is not expressed in mouse intestinal organoids (Lindeboom et al., 2018), which are responsive to RSPO3 ΔTSP/BR HS20 (Figure 5). This is explained by the fact the HS20 engages HS chains and thus can bind multiple HSPGs. We directly show that HS20 can interact equally with both GPC3 and GPC4 in a purified binding assay and that this interaction requires HS chains (Figure 4A).

3) The authors speculate on an interesting point that the ability of the different R-spondins to signal is "dictated by their ability to signal through HSPGs, LGRs, or both." Is there evidence that RSPO3 can interact with HSPGs and LGRs simultaneously? Perhaps examining whether the RSPO3-ΔTSP/BR-HS30 fusion protein can simultaneously interact with LGR4/5/6 and its known target GPC3 could help address this?

We have addressed this question by mutating the FU1 and FU2 domains of RSPO3 ΔTSP/BR HS20 individually to impair its interactions with ZNRF3/RNF43 and LGRs, respectively. As shown in two new figure panels (Figure 2G and H and associated discussion), mutating the LGR-interacting FU2 domain of RSPO3 ΔTSP/BR HS20 impairs its ability to potentiate WNT signaling in wild-type HAP1-7TGP cells, as does impairing the HSPG-interacting TSP/BR domain (Figure 1D). Notably, we had shown the same result for wild-type RSPO3 in our precursor *eLife* paper (Lebensohn and Rohatgi, 2018). These results demonstrate that LGRs and HSPGs enhance signaling by both RSPO3 ΔTSP/BR HS20 and wild-type RSPO3, consistent with the model that these ligands can engage LGRs and HSPGs simultaneously. However, as we note in the text, definitive evidence of a ternary complex will require additional biochemical and structural studies that are best suited for a dedicated future publication.

4) Given the inherent difficulties of interpreting negative results, the negative results Figure 4 may be minimized and presented as a figure supplement.

We agree that negative results can be difficult to interpret and explicitly note in the Discussion that our screen in one cell line does not preclude the existence of other RSPO receptors. However, the high-quality screen we present in Figure 6 (previously Figure 4) is the first unbiased, comprehensive genetic interrogation of LGR-independent signaling in a system where all three LGRs have been knocked out. Thus, the results of the screen will be of general interest to investigators working in this area, beyond the more focused question of the identity of an alternative RSPO receptor. We have chosen to keep the screen as a main figure element, so that it remains maximally visible (and thus usable) to readers. However, in response to this comment, we have significantly shortened the discussion around the screen and moved many of the details to the Materials and methods section.